# Impact of Isolation Procedures on the Development of a Preclinical Synovial Fibroblasts/Macrophages in an In Vitro Model of Osteoarthritis

**DOI:** 10.3390/biology9120459

**Published:** 2020-12-10

**Authors:** Cristina Manferdini, Yasmin Saleh, Paolo Dolzani, Elena Gabusi, Diego Trucco, Giuseppe Filardo, Gina Lisignoli

**Affiliations:** 1SC Laboratorio di Immunoreumatologia e RigenerazioneTissutale, IRCCS IstitutoOrtopedico Rizzoli, 40136 Bologna, Italy; cristina.manferdini@ior.it (C.M.); yasmin.saleh@ior.it (Y.S.); paolo.dolzani@ior.it (P.D.); elena.gabusi@ior.it (E.G.); diego.trucco@ior.it (D.T.); 2The BioRobotics Institute, Scuola Superiore Sant’Anna, 56127 Pisa, Italy; 3Department of Excellence in Robotics & AI, ScuolaSuperioreSant’Anna, 56127 Pisa, Italy; 4Applied and Translational Research (ATR) Center, IRCCS IstitutoOrtopedico Rizzoli, 40136 Bologna, Italy; giuseppe.filardo@ior.it

**Keywords:** synovial cells, synovial macrophages, synovial fibroblasts, isolation methods, osteoarthritis

## Abstract

**Simple Summary:**

In vitro models able to represent osteoarthritis (OA) synovial tissue (ST) inflammation are lacking. Therefore, we aimed to characterize OA ST and to compare mechanical and enzymatic digestion procedures to find the one that better preserve the heterogeneity of the main OA synovial cell populations: fibroblasts and macrophages. We demonstrated that typical macrophage phenotypical markers, like CD68, CD80 and CD163, were higher expressed on cells isolated with mechanical than enzymatic procedure. Moreover, we found that typical cytokines of inflammatory macrophages (i.e., TNFα) and anti-inflammatory macrophages (i.e., IL10) were also higher on mechanically isolated cells. Synovial fibroblasts were well conserved in both procedures. The definition of an OA ST model in vitro that better preserve the heterogeneity of specific cell populations represents a prerequisite for defining the real effects of new cell therapies or drugs for OA treatment, and could contribute to the reduction or avoidance of animal models.

**Abstract:**

There is a lack ofin vitromodels able to properly represent osteoarthritis (OA) synovial tissue (ST). We aimed to characterize OA ST and to investigate whether a mechanical or enzymatic digestion procedures influence synovial cell functional heterogeneity in vitro. Procedures using mechanical nondigested fragments (NDF), synovial digested fragments (SDF), and filtrated synovial digested cells (SDC) were compared. An immunophenotypic profile was performed to distinguish synovial fibroblasts (CD55, CD73, CD90, CD106), macrophages (CD14, CD68), M1-like (CD80, CD86), and M2-like (CD163, CD206) synovial macrophages. Pro-inflammatory (interleukin 6 IL6), tumor necrosis factor alpha (TNFα), chemokine C-C motif ligand 3 (CCL3/MIP1α), C-X- motif chemokine ligand 10 (CXCL10/IP10) and anti-inflammatory (interleukin 10 (IL10)), transforming growth factor beta 1 (TGFβ1), C-C motif chemokine ligand 18 (CCL18) cytokines were evaluated. CD68 and CD163 markers were higher in NDF and SDF compared to the SDC procedure, while CD80, CD86, and CD206 were higher only in NDF compared to the SDC procedure. Synovial fibroblast markers showed similar percentages. TNFα, CCL3/MIP1α, CXCL10/IP10, and CCL18 were higher in NDF compared to SDC, but not compared to SDF. IL10 and TGFβ1 were higher in NDF than SDC at the molecular level, while IL6 did not show differences among procedures. We demonstrated that NDF isolation procedures better preserved the heterogeneity of specific OA synovial populations (fibroblasts, macrophages), fostering their use for testing new cell therapies or drugs for OA, reducing or avoiding the use of animal models.

## 1. Introduction

Synovial inflammation is a key feature of osteoarthritis (OA), as well as of other arthritis conditions (such as rheumatoid arthritis, spondyloarthritis) [1,2]. It is involved in the development and progression of OA both in the early and late stages of the disease [3,4]. Synovial inflammation creates an unfavorable environment for the maintenance of a balanced joint tissues metabolism, which affects the capacity to counteract and repair damages [5,6].Several studies using ultrasound and magnetic resonance imaging, as well as histological techniques, have reported that in all stages of OA, there is a high prevalence of synovitis associated with pain and disability. This often represents the single driver of the catabolic processes in OA [1,7]. Synovial inflammation is characterized by increases in synovial membrane thickness and cell infiltration, mainly consisting of macrophages and lymphocytes and by the production of inflammatory mediators [8,9].

Macrophages are mainly located in the synovial lining layer and their number significantly increases from low to high grade synovial inflammation [2,7,10]. The primary role of macrophages is to maintain tissue homeostasis and protect the host from infection [11].Macrophages are classified as classically activated M1-like macrophages or alternatively activated M2-like macrophages depending on their polarization state in response to microenvironment stimuli [12,13]. M1-like macrophages are positive to CD80 and CD86 markers, secrete a high amount of pro-inflammatory cytokines, such as tumor necrosis factor alpha (TNFα), interleukin 1 beta (IL1β), interleukin 6 (IL6), C-X- motif chemokine ligand 10 (CXCL10/IP10), and a low level of IL10 [12,14,15]. M2-like macrophages are characterized by the expression of CD163 and CD206 markers, and by the production of IL10, C-C motif chemokine ligand 18 (CCL18), and transforming growth factor beta 1(TGFβ1) [12,16]. The deregulated balance between M1- and M2-like macrophages may lead to chronic low-grade inflammation [17]. The ratio of M1/M2 macrophages correlated positively with the Kellgren–Lawrence radiologic grading in OA patients [18,19]. Moreover, it has been shown that the complete depletion of M1 and M2 macrophages does not attenuate OA progression [18].

Therefore, understanding the dual role of macrophages in driving and resolving joint inflammation is a mandatory step in advancing OA management [20,21] and defining released factors as potential diagnostic biomarkers for OA [21,22]. Synovial tissue (ST) analysis is fundamental for clinical trial evaluations, translational science studies, and even in the routine clinical practice [23,24]. Different protocols for studying ST or isolating synovial cells (such as biopsy, immunohistochemistry, flow cytometry, molecular biology, in vitro cells, ex vivo explants, and cell lines) are available. However, the protocols used in the different studies could have positively or negatively influenced the results obtained [23,24,25,26,27,28,29,30].

Different studies focused onin vitro synovial fibroblast cell monolayers or on cell lines (i.e., K4IM), but both these models do not take into consideration the role of synovial macrophages or extracellular matrix (ECM) [27,31]. Other studies have used synovial explants to obtain the benefit of the native synovial membrane structure, but they have shown high variability and a they are limited by the low number of samples [32,33]. Up until now, there is a lack of in vitro systems that adequately represent the pathological ST. Although several attempts have been made to overcome these variations, an optimal strategy has not been identified yet.

To overcome this challenge, in this study we investigated whether different isolation procedures (mechanical separation versus enzymatic digested method) influence cell functional heterogeneity among synovial isolated cells. We proposed a protocol that permits the study of fresh human synovial cell populations and demonstrated that this method is useful in obtaining synovial fibroblasts and macrophages that adequately represent the pathological ST in vitro.

## 2. Materials and Methods

### 2.1. Patients Characterization

ST was obtained from 15 OA patients undergoing total knee replacement surgery, selected through clinical and radiological examinations (10 women and 5 men; mean age: 68 ± 6 years; body mass index: 28 ± 4 kg/m^2^; disease duration: 8 ± 3 years; Kellgren/Lawrence grade 3/4 [34]). The study was approved by the Rizzoli Orthopaedic Institute ethical committee and all patients provided their informed consent (Protocol number 15,274; Approval date: 12 April 2010).

### 2.2. Synovial Tissues Characterization

ST specimens were collected from at least two different areas, out of approximately 15 selected areas that showed macroscopically inflamed characteristics (like hypervascularity and hyperplasia) and were fixed in 4% formalin at room temperature for 16 h and then embedded in paraffin. Serial tissue specimen sections (4 µm thick) of each sample were prepared and routinely stained with hematoxylin-eosin. The histopathological features of each ST specimen were evaluated according to the synovitis scoring system described by Krenn [35]. This score system ranks with a scale from 0 to 3 each of the following parameters: hyperplasia of synovial lining layer, inflammatory infiltrate, and stromal cell density. The synovitis scoring system was summarized as follows: 0–1 no synovitis; 2–3 low-grade synovitis; 4–6 moderate-grade synovitis; and 7–9 high-grade synovitis. The scoring was performed by two independent observers (Cristina Manferdini and Paolo Dolzani).

### 2.3. Immunohistochemical Analysis of Synovial Tissue

Serial sections were deparaffinized, baked at 95 °C in epitope retrieval solution for 5–20 min (Target Retrieval Solution, Dako, Glostrup, Denmark) and incubated for 10 min at room temperature with Protein Block Serum-Free (Dako) to inhibit nonspecific staining. Then, slides were incubated for 1.5 h at room temperature with monoclonal anti-human-CD55 (10μg/mL Chemicon International Inc., Temecula, CA, USA), -CD68 (5μg/mL, Dako), -CD86 (10μg/mL GeneTex Inc., Irvine, CA, USA), and -CD206 (5μg/mL Abcam, Cambridge, UK) diluted in Tris-buffered saline (TBS) containing 0.1% bovine serum albumin (BSA). CD55 and CD68 single staining signals were developed with a biotin/streptavidin amplified, alkaline phosphatase-based, detection system (4+ Universal AP detection kit, Biocare Medical, Pacheco, CA, USA), using Fast Red as a substrate (Vulcan Fast Red Chromogen Kit, BiocareMedical, Pacheco, CA, USA).

CD86/CD206 double staining was evaluated by a commercial kit (MACH 2, Biocare Medical), containing a mixture of anti-mouse, anti-rabbit labelled with either Horseradish Peroxidase (HRP) and Alkaline Phosphatase (AP), and using as substrates diaminobenzidine (DAB) (Betazoid DAB Chromogen Kit, Biocare Medical) and Fast Red (Biocare Medical).

After nuclear counterstaining with hematoxylin, sections were mounted with glycerol gel and analyzed by brightfield microscope (Nikon Eclipse 90i, Nikon Instruments Europe BV, Quezon City, Philippines). Negative controls were performed using isotype controls (Dako). Semiquantitative analysis of immunohistochemically stained slides were performed on 20 microscopic fields (×200 magnification) for each section. The analysis was performed using Red/Green/Blue (RGB) with Software NIS-Elements 5.21.00 and Eclipse 90i microscope (Nikon Instruments Europe BV). Briefly, we acquired the total number of blue-stained nuclei and the total number of positive-stained red/brown cells. The data were expressed as percentage of positive cells for CD55, CD68, CD86, and CD206, respectively.

### 2.4. Synovial Cell Isolation by Using Both Mechanical and Enzymatic Procedures

To select synovial homogeneous tissue containing the most abundant cell populations (macrophages and fibroblasts) in sufficient amounts, we isolated synovial cells following both mechanical and enzymatic procedures, as summarized in Figure 1. Briefly, STs (2464 ± 285 mg/sample media ± SD) from approximately 13 different areas were cut into small fragments (approximately 2 mm^3^/each) with surgical forceps or scissors and equally divided into two parts for: mechanical nondigested fragments (NDF) cell isolation, a standardized procedure previously described [10], and enzymatic cell isolation, using collagenase IV from *Clostridium histoliticum* (400 U/mL) (Sigma-Aldrich, St Louis, MO, USA) for 1 h at 37°C in slow rotation. At the end of the enzymatic digestion, synovial digested cells (SDC) (at the bottom of the tube) were separated from synovial digested fragments (SDF) using a 70 µm cell strainer (BD, Franklin Lakes, NJ, USA). NDF, SDC, and SDF were cultured in petri dishes(Costar, Corning, NY, USA) and grown in OPTIMEM (Life Technologies Italia, Monza, Italy) supplemented with 100 U/mL penicillin, 100 µg/mL streptomycin, and 15% FBS (Euroclone, Pero, Italy) in a humidified atmosphere at 37°C with 5% CO_2_. After 7 days, fragments were removed from NDF and SDF cultures and medium changed from all isolated cells. We performed (passage 0) all the evaluations indicated in Figure 1 at day 10 only on moderate grade Krenn-scored isolated cells, while the low-grade ones were discarded.

CD14-positive and-negative synoviocytes were also isolated from SDF isolated cells in 4 cases out of 8, using anti-CD14-coated magnetic beads (MiltenyiBiotec GmbH, Bergisch Gladbach, Germany), as previously described [15,20].

### 2.5. Characterization of Isolated Synovial Cells

Freshly isolated synovial cells from NDF, SDF, and SDC (passages 0) were detached at day 10 by trypsin (Life Technologies),counted with eosin (NDF: 0.98 × 10^6^ ± 1.5 × 10^5^; SDF: 1.1 × 10^6^ ± 1.2 × 10^5^; SDC: 1.2 × 10^6^ ± 1.1 × 10^5^, cells were 99% live), and characterized by flow cytometry using markers expressed by synovial fibroblasts (SF): monoclonal anti-human-CD55 (2.5 μg/mL, Chemicon), -CD73(5 μg/mL, BD), -CD90 (5 μg/mL, BD), -CD106 (10 μg/mL, Merck Millipore, Milano, Italy); by synovial macrophages (SM): monoclonal anti-human-CD14 (5 μg/mL, Dako), -CD68 (5 μg/mL, Dako), -CD80 (5 μg/mL, GeneTex Inc.), -CD163 (10 μg/mL, Abcam, Cambridge, UK), -CD86 (5 μg/mL, GeneTex Inc.), -CD206 (5 μg/mL, Abcam), as well as by endothelial (CD31, 2 μg/mL, R&D System Inc., Minneapolis, MN, USA) and mononuclear cells (CD3, CD34, CD45RA) (5 μg/mL, Dako). Briefly, cells were harvested, washed twice with phosphate- buffer saline (PBS), centrifuged, washed in PBS supplemented with 2% BSA and 0.1% sodium azide (flow cytometry buffer), and aliquots of 1 × 10^5^ cells were then incubated with specific primary antibodies at 4°C for 30 min. Cells were washed twice with a flow cytometry buffer and incubated with polyclonal rabbit anti-mouse immunoglobulins/FITC conjugate (DakoCytomation) at 4 °C for 30 min. After two final washes, cells were analyzed using a FACS CantoII Cytometer (BD). For isotype control, a nonspecific mouse IgG was substituted to the primary antibody.

### 2.6. Cytokine and Chemokine Release in Supernatants

The concentrations of IL6, CXCL10/IP10, IL10, CCL3/MIP1α, TNFα, CCL18, and TGFβ1 were simultaneously evaluated in the supernatants of 10^6^ synovial cells from NDF, SDF, and SDC isolation procedures (passages 0) using multiplex bead-based sandwich immunoassay kits (BioRad Laboratories Inc., Segrate, Italy) following the manufacturer’s instructions. Briefly, we added 50μL to each well of the diluted standards (fourfold dilution series), controls and samples in triplicate and added 50μL of coupled beads and the plate was incubated at room temperature for 30 min. The plate was then washed three times with 100μLwash buffer and incubated with 25 μL of detection antibodies for 30 min. Finally, the plate was washed three times and incubated with 50 μL of streptavidin-PE for 30 min and measured in a reader (Luminex Bio-plex System, Bio-Rad Laboratories Inc. Segrate, Italy).

### 2.7. Real Time-PCR Analysis

Total RNA was extracted from NDF, SDF, and SDC isolated human synovial cells (passages 0), using RNA PURE reagent (Euroclone Spa, Pero, Italy) according to the manufacturer’s instructions, and then treated with DNase I (DNA-free Kit, Life Technologies). Reverse transcription was performed using SuperScript VILO (Life Technologies) reverse transcriptase and random hexamers, following the manufacturer’s protocol. 

Forward and reverse oligonucleotides for PCR amplification of Glyceraldehyde 3-phosphate dehydrogenase (GAPDH), CD80, CD86, CD163, CD206, IL6,TNFα, CCL3/MIP1α, CXCL10/IP10, IL10, TGFβ1, and CCL18 are described in Appendix A and real time PCR was carried out as previously described [36]. All primer efficiencies were confirmed to be high (>90%) and comparable (Appendix A). For each target gene, mRNA levels were calculated, normalized to GAPDH according to the formula 2^−∆Ct^ and expressed as a percentage of the reference gene since this was expressed in the same amount in all conditions tested. 

### 2.8. Statistical Analysis

Statistical analysis was performed using nonparametric tests since the data did not have a normal distribution. Mann–Whitney U test was used to compare CD55 versus CD68 and CD14-positive versus CD14-negative cells. Friedman and Dunn’s post-hoc test was used to compare CD68, CD86, and CD206 or NDF, SDF, and SDC. GraphPad Prism 8 (GraphPad Software Inc., La Jolla, CA, USA) was used for the analysis, and values of *p* < 0.05 were considered statistically significant. Values were expressed as median, minimum and maximum.

## 3. Results

### 3.1. Characterization of OA Synovial Tissue Samples

ST from 15 patients with OA were first scored on hematoxylin-eosin-stained slides, as reported by Krenn [35], to select cases with moderate-grade synovitis that showed a high percentage of the two main representative populations of ST (fibroblasts and macrophages). We found low-grade synovitis in 7 samples (that were discarded) and moderate-grade synovitis in 8 samples (Figure 2A). Then, for an in-depth analysis of the main synovial cell populations present in the ST, we analyzed CD68 and CD55, both well-known markers of synovial macrophages and synovial fibroblasts. As shown in Figure 2B, negative control and in Figure 2C, CD68 was mainly positive on synovial macrophages located in the lining layer and on a few macrophages in the sublining layer. CD55 was positive both on the lining and sublining layers (Figure 2D), as previously reported [10].

To identify the distribution of different macrophage subsets in OA ST with moderate-grade synovitis, we evaluated the expression of CD86 and CD206 in 6 out of 8 different cases. As we previously demonstrated [15], we confirmed a characteristic distribution of CD86 as M1-like and CD206 as M2-like macrophage markers. Briefly, M1-like macrophages were mainly located in the lining layer while M2-like macrophages showed a peculiar distribution depending on the areas; in some areas they were located only in the sublining, while in other areas they were located both in the lining and sublining (Figure 2E,F). Moreover, a quantitative analysis of the percentage of positive cells to CD55, CD68, CD86, and CD206 was also performed. As shown in Figure 2G, we evidenced a significant (*p* < 0.0001) higher percentage of CD55 (mean ± SD64.4 ± 6.8) compared to CD68 positive cells (mean ± SD 24.7 ± 7.4) and we found a lower percentage of CD86 (mean ± SD 16.5 ± 6.2) and CD206 (mean ± SD 13.6 ± 5.7) compared to CD68-positive cells.

### 3.2. Synovial Cell Isolation Procedures Yield a Different Percentage of Macrophages

Isolated synovial cells from NDF, SDF, and SDC were evaluated by flow cytometry at passage 0. As shown in Figure 3, we found that synovial fibroblast markers (CD55, CD73, CD90, and CD106, selected based on previous reports [37,38,39]) showed a similar percentage of positive cells in all three procedures used. Interestingly, synovial macrophage marker CD14 showed the same percentage, while CD68 was significantly higher in NDF and SDF compared to SDC (*p* = 0.0424 and *p* = 0.0183, respectively). In all three procedures CD3, CD31, and CD45RA typical markers of T lymphocytes, endothelial and hematopoietic cells, showed a very low percentage (not more than 2%), while CD34 was absent.

### 3.3. Synovial Cell Isolation Procedures Yield a Different Expression of Synovial M1- and M2-Like Macrophage Markers

We then focused on typical markers of M1- and M2-like synovial macrophages CD80, CD86, CD163, and CD206 that were analyzed both at molecular and protein levels. As shown in Figure 4A–D, we demonstrated at the molecular level that both CD80, CD86, CD163, and CD206 were significantly higher (*p* = 0.0071, *p* = 0.0370, *p* = 0.014, *p* = 0.014, respectively) in NDF than in SDC isolated cells. Only CD163 was significantly higher in SDF compared to SDC (*p* = 0.04).

We confirmed by flow cytometry that CD80, CD86, CD163, and CD206 were also significantly higher, both as percentage of positive cells (*p* = 0.0397, *p* = 0.0483, *p* = 0.0436, *p* = 0.0035, respectively) and as mean fluorescent intensity (MFI, *p* = 0.0133, *p* = 0.0400, *p* = 0.0080, *p* = 0.0040, respectively) in NDF compared to SDC isolated cells (Figure 4E–I,L–N).

### 3.4. Synovial Cell Isolation Procedures Yield a Different Level of Synovial Cell Released Factors

We then evaluated typical factors released by synovial M1- and M2-like macrophages. As shown in Figure 5, typical factors released by M1-like macrophages, such as TNFα, CCL3/MIP1α, and CXCL10/IP10, were significantly higher in NDF compared to SDC both at molecular (*p* = 0.0044, *p* = 0.0175, *p* = 0.0097, respectively) and protein levels (*p* = 0.0011, *p* = 0.0023, *p* = 0.009, respectively), while IL6 did not show any difference.

Interestingly, also typically released factors of M2-like macrophages, such as IL10, TGFβ1, and CCL18, were significantly higher in NDF compared to SDC at molecular (*p* = 0.011, *p* = 0.0486, *p* = 0.0140, respectively) and only CCL18 at protein levels (*p* = 0.0044) (Figure 6). TGFβ1 was significantly higher in NDF than in SDF (*p* = 0.006) at the molecular level.

### 3.5. Lower Amounts of All Measured Secreted Factors in Both CD14-Positive and CD14-Negative Populations

Finally, the same M1- and M2-like macrophages released factors were analyzed on a pure population of both CD14-positive and negative synoviocytes to define the differences with mixed cell populations of isolated NDF, SDF, and SDC. To verify the purity of the CD14-positive and negative synoviocytes, we tested with flow cytometry also other macrophage markers (CD68, CD86 and CD206) and we confirmed their expression only on CD14-positive synoviocytes (Appendix A). As shown in Figure 7, pure CD14-positive synoviocytes expressed (Figure 7) significantly higher amounts of TNFα, CCL3/MIP1α, CXCL10/IP10, and CCL18 (*p* = 0.0043, *p* = 0.0043, *p* = 0.0022, *p* = 0.0043) compared to CD14-negative cells. As shown in Figure 8, we confirmed that pure CD14-positive synoviocytes also released a significantly higher amount of TNFα, CCL3/MIP1α, CXCL10/IP10, and CCL18 (*p* = 0.0079, *p* = 0.0006, *p* = 0.0159, *p* = 0.0286) compared to CD14-negativesynoviocytes, but lower than NDF or SDF (Figure 5 and Figure 6). Moreover, IL6, IL10, and TGFβ1 showed similar amounts in both CD14-positive and negative cells both at molecular (Figure 7) and protein (Figure 8) levels, as well as in NDF, SDF, and SDC (Figure 5 and Figure 6).

## 4. Discussion

ST from different diseases is widely studied by histological or immunohistochemical evaluations or by focusing on isolated cells [24]. Synovitis represents a key feature in OA [1,3] and the definition of synovial isolation procedures that consider the characteristics of native ST could better allow the validation of the data obtained using 2D cell cultures [40]. Thorough analysis of the histological characteristics of ST showed that it is composed of different cell types. Synovial macrophages and fibroblasts represent the highest percentage of cells followed by mononuclear and endothelial cells [41]. The isolation of synovial cells is performed using procedures mainly based on different enzymatic treatments. To our knowledge, only two recent studies have focused on disaggregation conditions to isolate OA and RA synovial cells focusing on flow cytometry and multiplex data analysis [27], and on mass cytometry and RNA-seq transcriptome analysis [26]. In our study, we compared a mechanical non enzymatic procedure with a collagenase-based enzymatic procedure to isolate OA synovial cells from previously characterized STs by Krenn immunological synovitis score [42].

First, we defined the grade of synovitis of the OA samples analyzed, and then selected only samples with Krenn score moderate-grade synovitis that had a higher percentage of both synovial fibroblasts and macrophages. These scored synovial samples were used to isolate cells using both mechanical and enzymatic procedures. Our data demonstrated that the two synovial isolation procedures did not differently affect (at passage 0) the percentage of positive isolated cells on typical markers of synovial fibroblasts (CD55, CD73, CD90, CD106), suggesting that both mechanical or enzymatic treatments had no specific impact on surface markers, confirming a high percentage of isolated synovial fibroblasts compared to macrophages [43]. However, for typical macrophage markers we did not find differences using the CD14 monoclonal antibody, but using CD68, CD80, or CD86 we found that the SDC procedure had a lower cell percentage than NDF or SDF. Interestingly, the percentage of synovial fibroblasts and macrophages that we quantified on ST samples with moderate synovitis was similar to the percentage of the isolated cells, confirming that both cell populations were well represented in vitro. All the isolation procedures used did not affect CD14, even though it has been described as being a marker sensitive to enzymatic treatment [44]. However, these procedures did not permit the isolation of mononuclear (T lymphocytes, hematopoietic) and endothelial cells that were found in very low percentage or were completely absent. In line with our data, it has been shown [26,27] that different enzymatic digestions used to isolate synovial cells negatively impacted the isolation of mononuclear cells from ST. Still, the mechanical procedure used did not help to recover endothelial and mononuclear cells from ST. Thus, the lack of these cells after the isolation procedures seem independent from the use of collagenase IV-based enzymatic digestion. On the other hand, the lack of these cells could also be dependent on the focus of the studies only on adherent cells and not on cells in suspension (non adherent).

It has been shown on fresh PBMC [23,26,27] that enzymatic digestion influences the detection of immune cells markers, which could corroborate our results on retained immune adherent cells. Furthermore, CD68, another typical marker of macrophages, showed a significant increase in NDF and SDF compared to SDC, suggesting that this cell population could be affected by the isolation procedure used and, as previously reported, by the passage in culture [10] or by the ability of macrophages to polarize their phenotype [45].

When we focused on typical markers of M1- or M2-like macrophages (CD80, CD86, CD163, and CD206), we observed a significant decrease of these cell populations in SDC compared to NDF. A similar trend was also observed with SDF, even if it did not reach the statistical significance at both molecular and protein levels, except for CD163. M1- and M2-like synovial macrophages seemed entrapped in the fragments and the enzymatic digestion partially contributed to facilitate their release in culture. In fact, the mechanically and enzymatically treated fragments showed no difference in the percentage of M1- and M2-like synovial macrophages, showing a similar recovery of these cells. However, the enzymatic procedure (SDF) could have the advantage of shortening the time necessary for the release of cells in culture.

The analysis of released factors typical of M1- and M2-like synovial macrophages from the same number of isolated cells evidenced, both at molecular and protein levels, that TNFα, CCL3/MIP1α, CXCL10/IP10 (M1 released factors), and CCL18 (M2 released factor) were significantly higher in NDF compared to SDC, but not different from SDF. All these factors were significantly higher also in pure populations of CD14-positive OA synovial cells compared to CD14-negative ones, confirming that they are mainly released by M1-like synovial macrophages. The same trend was also confirmed using NDF and SDF procedures, that better preserved M1- and M2-like synovial macrophages in culture, and showed a high ratio of M1/M2 as found in OA synovial fluid and peripheral blood [18]. Moreover, different studies [46,47,48] on OA synovium evidenced that these molecules are associated to synovial macrophages polarization and disease severity [2], confirming the importance of preserving these cell populations for in vitro studies. In fact, it has been shown that CCL3/MIP1α deficiency significantly protect mice from cartilage damage, synovitis, and cell infiltration [49]. Therefore, the detection of this chemokine after isolation procedures confirm the retention of key specific synovial cell populations in culture.

By contrast, we found that IL6, a typical cytokine produced by synovial M1-like macrophages [20], showed the same amount independently from the isolation procedure used. It has also been shown that OA synovial fibroblast in basal condition release IL6 [50]. Our data confirmed, on pure populations of CD14-positive and-negative OA synovial cells, that both macrophages and fibroblasts release a similar amount of this cytokine. Moreover, the amount of IL6 detected was independent from the contemporary presence of both synovial fibroblasts and macrophages, as found in NDF, SDF, and SDC isolated cells.

IL10 and TGFβ1 were significantly higher in NDF than SDC at molecular level but not at the protein level, even if the percentage of M2-like synovial macrophages (directly involved in the production of IL10 and TGFβ1) was significantly higher in NDF, but not different from SDF. Moreover, we also found that, both at molecular and protein levels, IL10 and TGFβ1 showed the same expression both in CD14-positive and negative OA synovial cells, and that IL10 was released in low amount as previously reported [14]. It is well known that TGF β1 and IL10 are both expressed by fibroblasts and macrophages and CD14+ and CD14− and are modulated during M2 polarization [12,51,52]. This suggests that these factors could be accumulated into the cells while maintaining a basal protein release and only specific conditions could modulate their release. In fact, it has been shown that high or low concentrations of TGFβ1 differently stimulate the balance among key signaling pathways (i.e., SMAD2/3 versus SMAD 1/5/8 [53,54]). IL10 in OA patients is produced by a subset of T [55] and B [56] lymphocytes, substantiating the results obtained in our study that found them in low percentage or absent.

TGFβ1 has been shown to be localized in the lining layer of OA ST [57] and produced by both macrophages [58] and fibroblasts [43], clearly indicating that both synovial cell types contribute to its production, in line with our results on both pure CD14-positive and negative cells, as well as on synovial cells isolated with the different procedures.

CCL18 was the only typical M2 macrophage cytokine that we found significantly higher in the NDF procedure compared to SDC, but not different than the SDF procedure, demonstrating that this cell population was well preserved in culture. It has been shown that CCL18 is associated with OA disease severity and directly correlated with radiographic grading [59], suggesting that M2-like macrophages contribute to the evolution of the OA disease. Therefore, the maintenance of M2-like macrophages in culture is an important requisite to better the mimic OA milieu condition.

The limit of this study is the number of collected synovial samples from each patient that influenced the total amount of cells obtained for performing the experiments. By contrast, the strength of this study is based on the use of a mixed population of synovial fibroblasts and macrophages. This avoids the use of a restricted number of synovial explants, which do not adequately represent the overall composition of the OA ST (characterized by patched inflamed areas) and limits the evaluation at different experimental points. Moreover, the mixed cell populations model could also contribute to defining the interactions and signaling among the cells. We have previously demonstrated [10] that low inflamed cultures of synovial fibroblasts and macrophages are not efficient for testing cell OA therapies, and also the use of synovial cell monocultures could represent a limit to reproduce an OA-like environment in vitro. The development of a 3D synovial structure combining synovial cells isolated with these procedures with scaffolds biomimetic for the extracellular matrix could open new directions that overcome the limits of 3D synovial cells micromasses [60]. Finally, these results corroborate our previous papers [10,61] that demonstrated the effect of adipose derived stromal cells (as OA cell therapies) on synoviocytes isolated with the NDF procedure.

## 5. Conclusions

This study underlines the importance of defining the correct synovial cell isolation procedure to obtain an in vitro cell culture model that adequately preserves the cell populations and retains the features of synovial pathological native tissue. The NDF isolation procedure can retain synovial fibroblasts and a high ratio of M1/M2-like macrophages, as we found in ST, suggesting that an OA-like environment was maintained in the in vitro culture, and providing new indications and future directions for synovial OA research in vitro models. The definition of an in vitro model approximate to the pathological condition represents a prerequisite for defining the real effects of new cell therapies or drugs for OA treatment, and could contribute to the reduction or avoidance of animal models.

## Figures and Tables

**Figure 1 biology-09-00459-f001:**
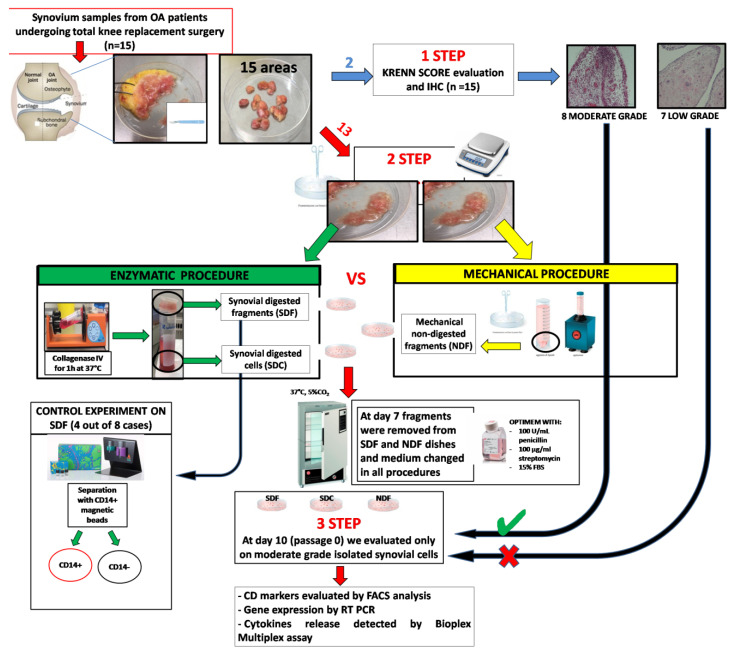
Experimental plan. Synovium samples were selected from 15 OA (osteoarthritis) patients undergoing total knee replacement. From each sample, approximately 15 areas were selected and 2 of them were analysed by histology for Krenn score evaluation (STEP 1) and immunohistochemistry (IHC). The other 13 areas were minced, weighted and equally divided for mechanical and enzymatic procedures (STEP 2). For the mechanical non digested fragments (NDF) cell isolation, pieces were further minced and gently washed. For enzymatic procedure, minced pieces were treated with collagenase IV for 1h at 37 °C to obtain synovial digested cells (SDC) (at the bottom of the tube) that were separated from synovial digested fragments (SDF) using a 70-µm cell strainer. NDF, SDC and SDF were cultured in petri dishes and grown in OPTIMEM with 100 U/mL penicillin, 100 µg/mL streptomycin and 15% FBS (Euroclone) in incubator. After 7 days, fragments were removed from NDF and SDF cultures and medium changed from all isolated cells. At day 10 (passage 0, STEP3) cells from low-grade scored ST were discarded and cells from moderate grade scored ST were evaluated for CD markers, gene expression andreleased cytokines. Control experiments were performed on a pure population of CD14-positive (+) and -negative (−) synoviocytes isolated from SDF.

**Figure 2 biology-09-00459-f002:**
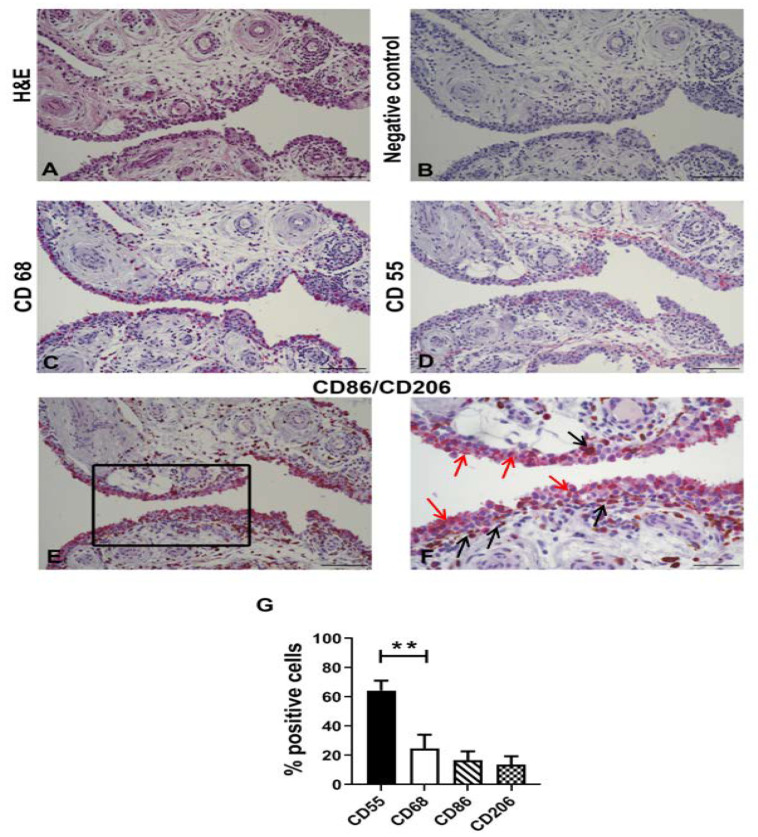
Characterization of moderate grade synovial tissue samples. (**A**) Representative image of Hematoxylin-Eosin (H&E) stained synovial tissue with moderate grade synovitis for Krenn score analysis. (**B**–**F**) Representative images of CD68 (**C**), CD55 (**D**), CD86/CD206 (**E**,**F**) shows a high magnification image of the square area indicated in (**E**), bar = 50 µm, red arrows indicate CD86 and black arrows CD206) immunohistochemical analysis on synovial tissue with moderate grade synovitis. Positive stained cells in red (CD68, CD55, CD86) or brown (CD206). Negative control (**B**). Bars = 100 µm. (**G**) Percentage of positive cells to CD55, CD68, CD80, and CD206 analyzed in moderate-grade synovitis in OA (*n* = 8). Data are expressed as percentage of positive cells and represented as box plots with mean ± SD. Significant results among groups (** *p* < 0.01) were reported (Mann–Whitney U test was used to compare CD55 versus CD68).

**Figure 3 biology-09-00459-f003:**
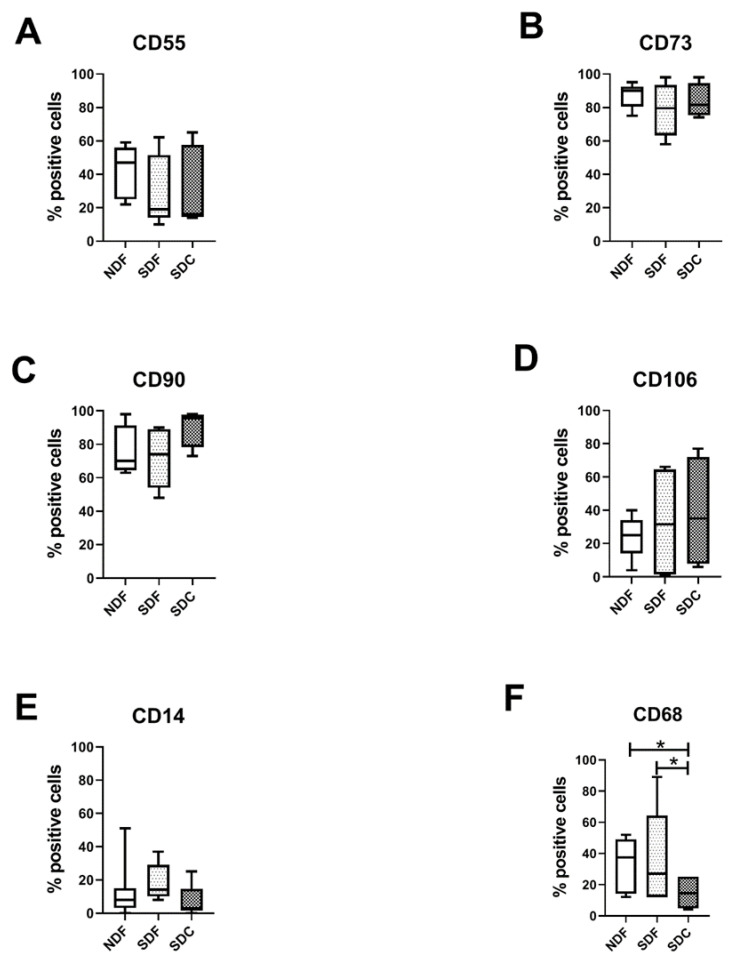
Flow cytometry evaluation of markers of synovial fibroblasts and macrophages. CD55 (**A**), CD73 (**B**), CD90 (**C**), and CD106 (**D**) typical markers of synovial fibroblasts evaluated on cells obtained with NDF, SDF, and SDC isolation procedures (*n* = 8). CD14 (**E**) and CD68 (**F**) typical markers of synovial macrophages evaluated on cells obtained with NDF, SDF, and SDC isolation procedures. Data are expressed as percentage of positive cells and represented as box plots with median, minimum and maximum. Significant results among groups (* *p* < 0.05) were reported (Dunn’s for multiple comparisons test).

**Figure 4 biology-09-00459-f004:**
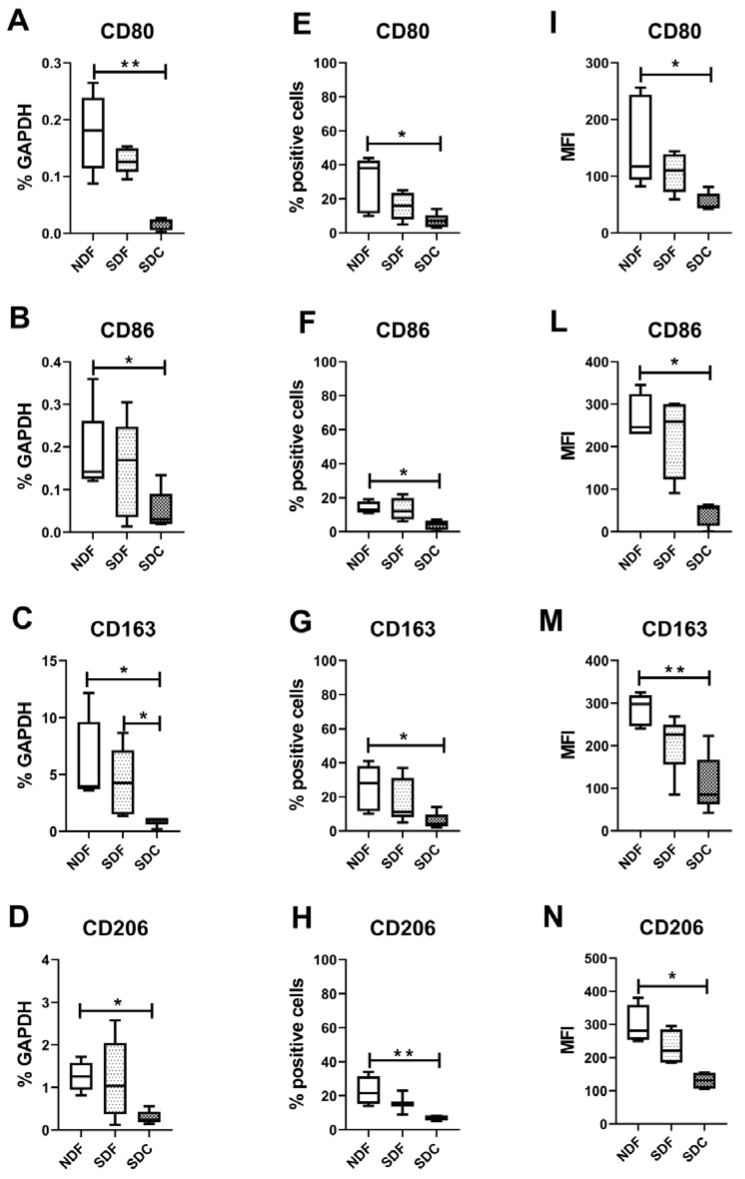
Evaluation of markers of synovial M1- and M2-like macrophages. CD80 (**A**,**E**,**I**), CD86 (**B**,**F**,**L**) typical markers of M1-type macrophages evaluated at both molecular (**A**,**B**) and protein (**E**,**F**,**I**,**L**) levels on cells obtained with NDF, SDF, and SDC isolation procedures (*n* = 8). CD163 (**C**,**G**,**M**), CD206 (**D**,**H**,**N**) typical markers of M2-type macrophages evaluated at both molecular (**C**,**D**) and protein (**G**,**H**,**M**,**N**) levels on cells obtained with NDF, SDF, and SDC isolation procedures. Data are expressed as percentage of GAPDH (housekeeping gene) or as percentage of positive cells, or as mean fluorescent intensity (MFI), and represented as box plots with median, minimum and maximum. Significant results among groups (* *p* < 0.05, ** *p* < 0.01) were reported (Dunn’s for multiple comparisons test).

**Figure 5 biology-09-00459-f005:**
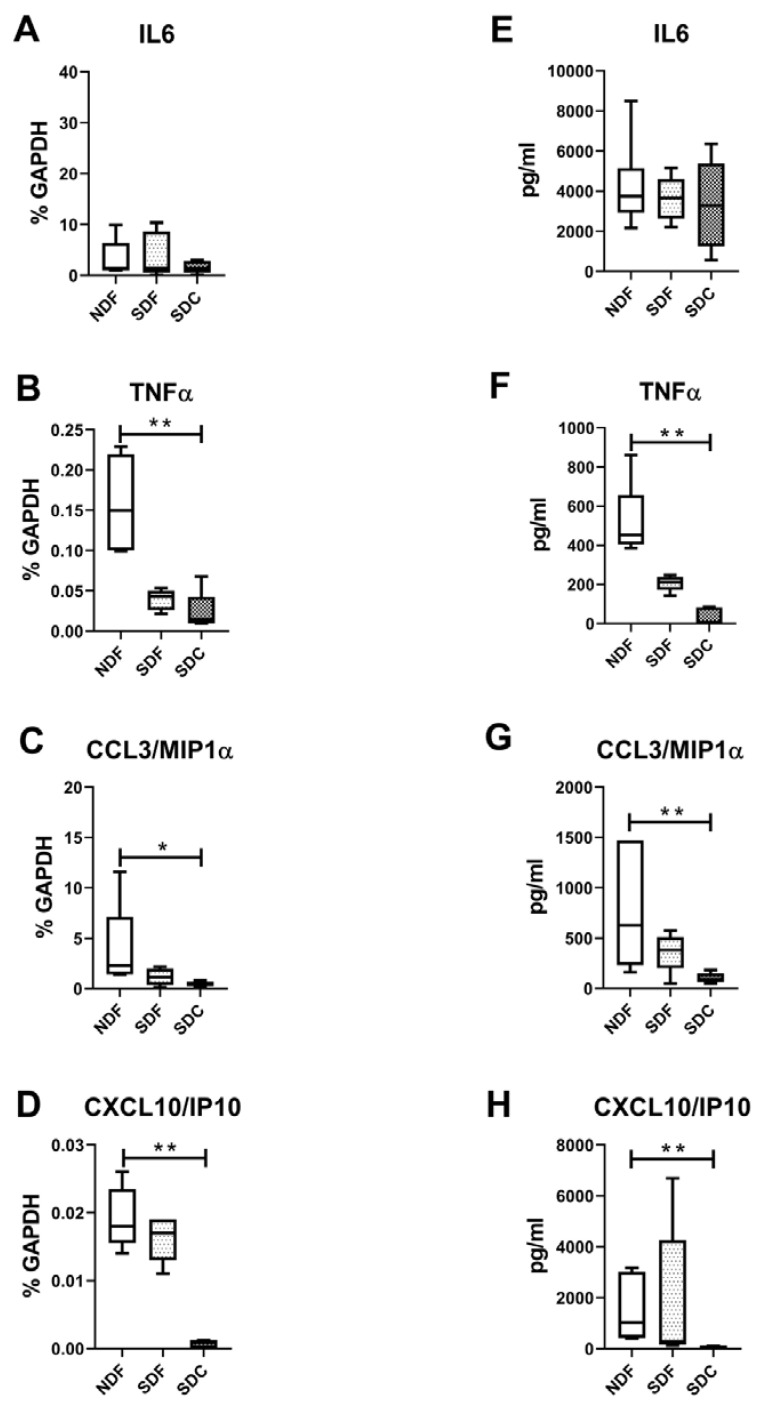
Evaluation of cytokines typical of synovial M1-like macrophages. IL6 (**A**–**E**), TNFα (**B**–**F**), CCL3/MIP1α (**C**–**G**), CXCL10/IP10 (**D**–**H**) typical cytokines of M1-type macrophages evaluated at both molecular (**A**–**D**) and protein (**E**–**H**) levels on cells obtained with NDF, SDF, and SDC isolation procedures (*n* = 8). Data are expressed as percentage of Glyceraldehyde 3-phosphate dehydrogenase GAPDH (housekeeping gene) or as pg/mL and represented as box plots with median, minimum and maximum. Significant results among groups (* *p* < 0.05, ** *p* < 0.01) were reported (Dunn’s for multiple comparisons test).

**Figure 6 biology-09-00459-f006:**
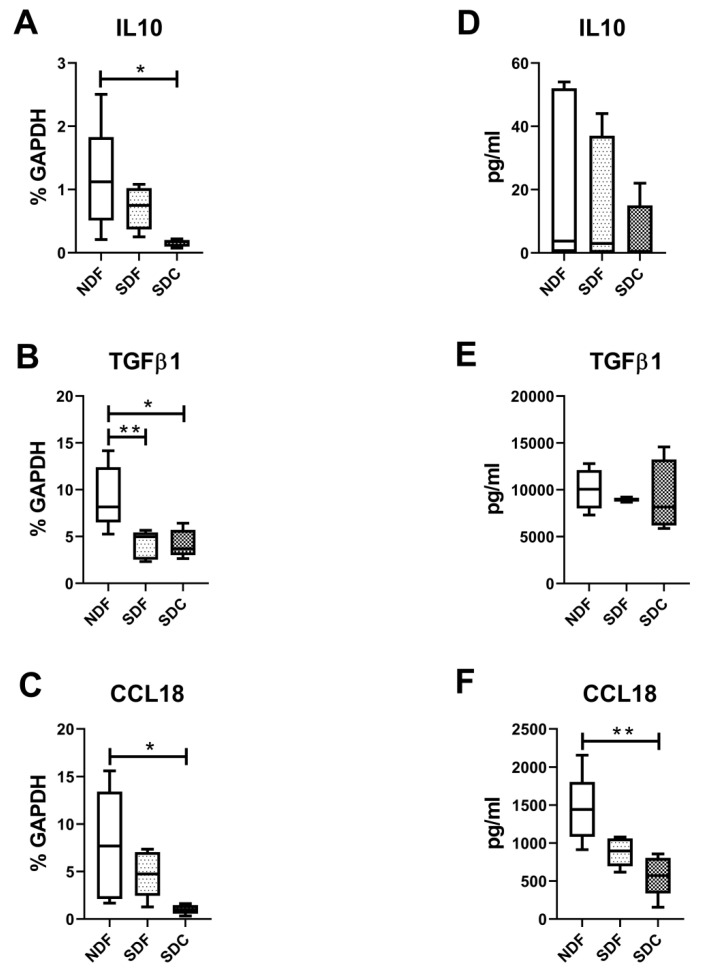
Evaluation of cytokines typical of synovial M2-like macrophages. IL10 (**A**–**D**), TGFβ1 (**B**–**E**), CCL18 (**C**–**F**) typical cytokines of M2-type macrophages evaluated at both molecular (**A**–**C**) and protein (**D**–**F**) levels on cells obtained with NDF, SDF, and SDC isolation procedures (*n* = 8). Data are expressed as percentage of GAPDH (housekeeping gene) or as pg/mL and represented as box plots with median, minimum and maximum. Significant results among groups (* *p* < 0.05, ** *p* < 0.01) were reported (Dunn’s for multiple comparisons test).

**Figure 7 biology-09-00459-f007:**
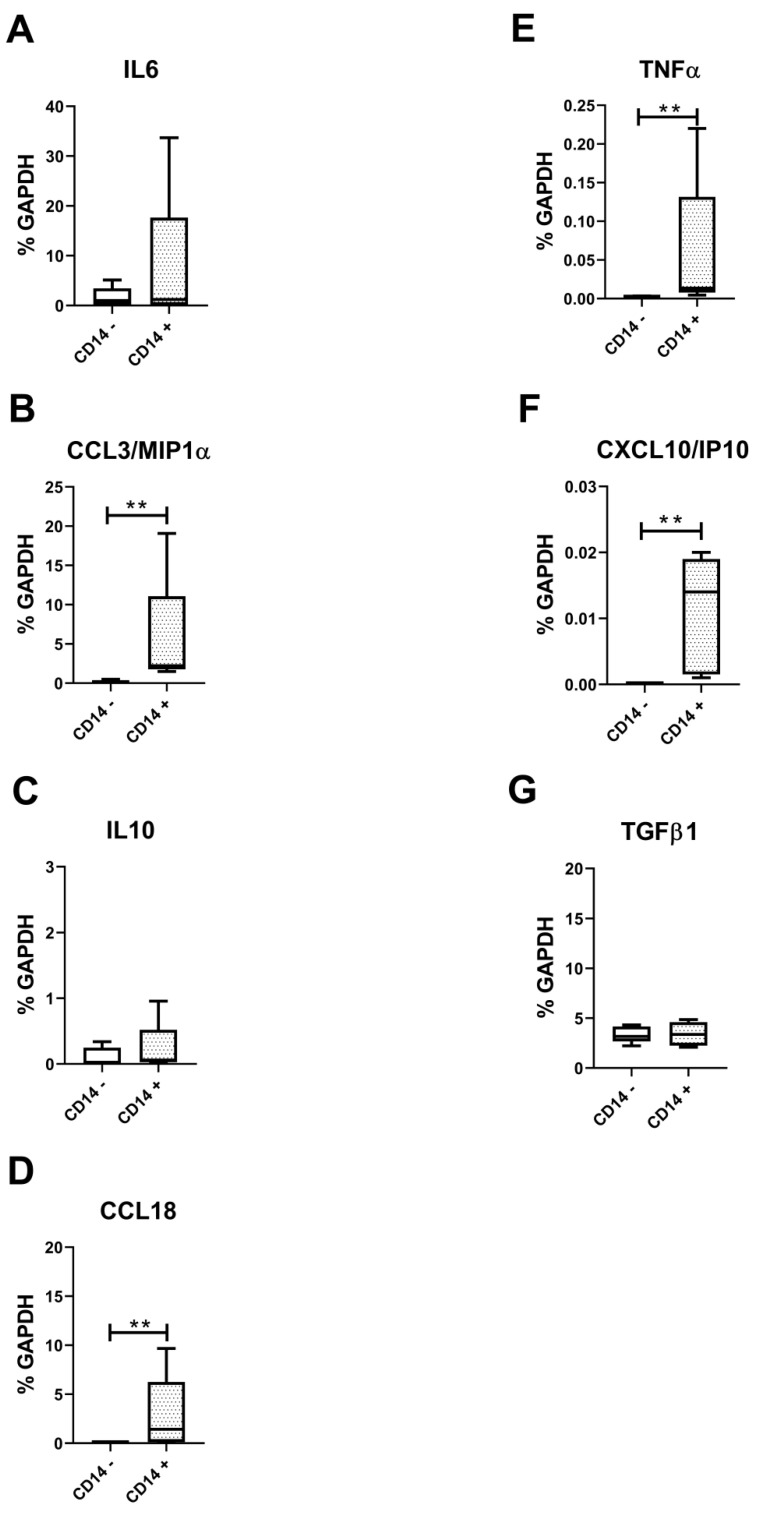
Evaluation of cytokines typical of synovial M1- and M2-like macrophages. (**A**) IL6, (**B**) CCL3/MIP1α, (**C**) IL10, (**D**) CCL18, (**E**) TNFα, (**F**) CXCL10/IP10, (**G**) TGFβ1 typical cytokines of M1 and M2-type macrophages evaluated at the molecular level on pure CD14 negative (CD14−) and CD14-positive (CD14+) cells isolated from OA synovium (*n* = 4). Data are expressed as percentage of GAPDH (housekeeping gene) and represented as box plots with median, minimum and maximum. Significant results among groups (** *p* < 0.01) were reported (Mann–Whitney U test was used to compare the two groups).

**Figure 8 biology-09-00459-f008:**
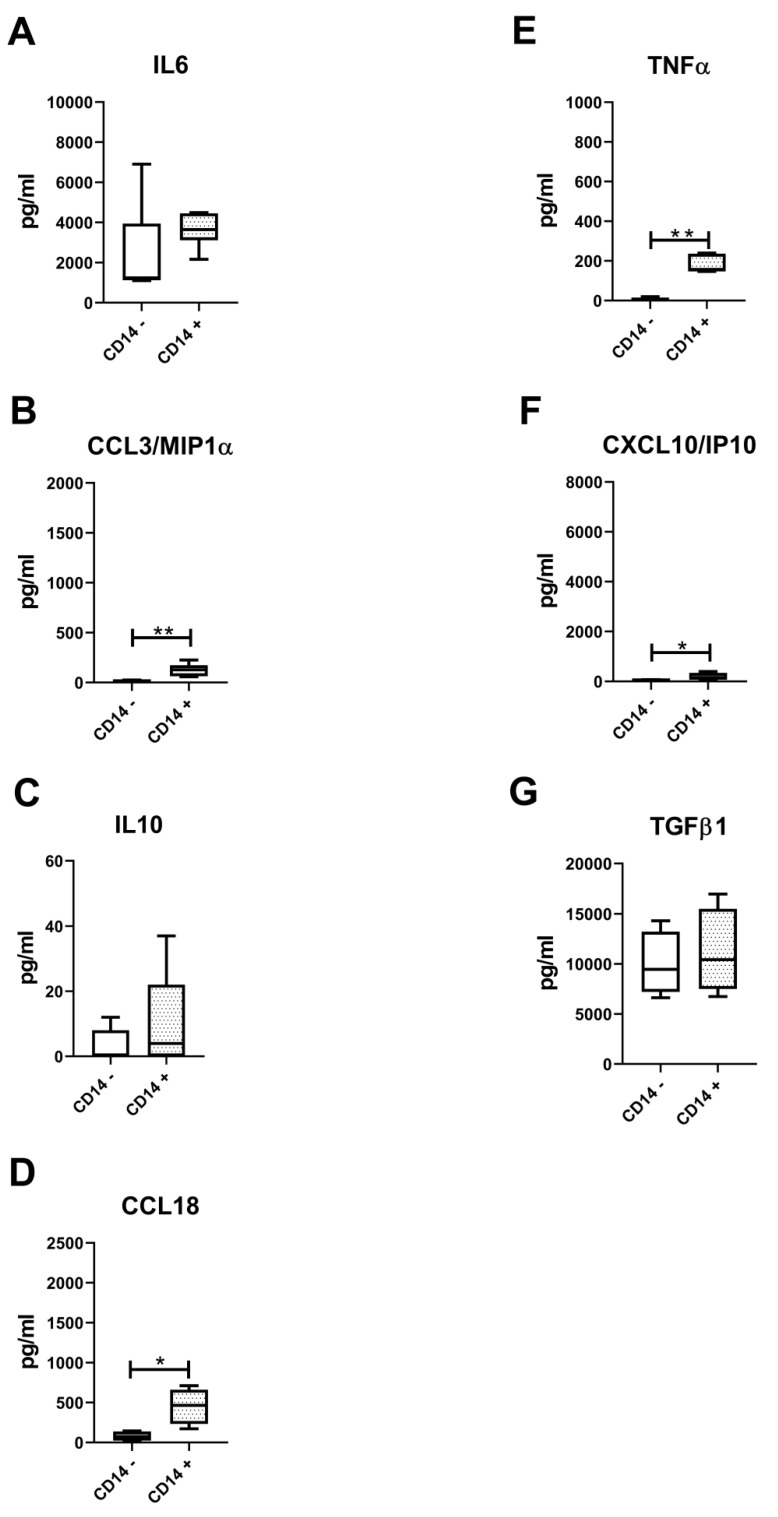
Evaluation of cytokines typical of synovial M1- and M2-like macrophages. (**A**) IL6, (**B**) CCL3/MIP1α, (**C**) IL10, (**D**) CCL18, (**E**) TNFα, (**F**) CXCL10/IP10, (**G**) TGFβ1 typical cytokines of M1 and M2-type macrophages evaluated at the protein level on pure CD14 negative (CD14−) and CD14-positive (CD14+) cells isolated from OA synovium (*n* = 4). Data are expressed as pg/mL and represented as box plots with median, minimum and maximum. Significant results among groups (* *p* < 0.05, ** *p* < 0.01) were reported (Mann–Whitney U test was used to compare the two groups).

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
