# Peer review of "Impact of Isolation Procedures on the Development of a Preclinical Synovial Fibroblasts/Macrophages in an In Vitro Model of Osteoarthritis"

_biology, 2020, doi:10.3390/biology9120459_

Round 1

Reviewer 1 Report

There seems to be a lot of data that was collected for this study, but a lot of the analysis did not support the conclusions made. Results and discussion should focus on a more logical step-by-step to break down the synovial cell populations observed. There is a interchangeable use of what constitutes as your synovial fibroblast and synovial macrophage markers without ample justification, particularly where two markers are both claimed as well-known markers, but do not represent positive populations of each other.  

Line-specific comments: 

68-71: Some of these references are not specific to OA. Is this of concern? Other arthritic conditions have different manifestations of synovium inflammation and tissue condition that may call for different isolation techniques.  

71: Multiple publications are cited here. What kind of variation was seen in these protocols and what made the variation ‘significant’?  

81-84: Have you completed validation experiments to support the claim that the isolated cell populations can be used to test new therapies for OA? The current data presented does not show if the three different isolation protocols produce cells that respond differentially to known pro- or anti-inflammatory stimuli. Perhaps further elaboration or citation of previous work would be sufficient.  

95-96: How were ‘inflamed characteristics’ selected from the synovial tissue specimens for fixation? Is this done by the surgeon or the lab operator? What characteristics defined this selection before Krenn scoring was done on the H&E samples?  

Figure 1: This current experimental schematic is unclear. Was the mechanical separation completed on the enzymatic digestion portion? In the Methods, the tissue was described to be cut into small fragments before splitting into ‘mechanical separation’ vs ‘enzymatic digestion’, but the schematic appears to place the scissors only in the enzymatic digestion route.  

Figure 1: The 7-day culture of the fragments in petri dishes and subsequent 10-day culture should be outlined in this schematic before ‘evaluation performed’. Please clarify if all evaluation and harvest of the cell populations was only done after a total of 17 days incubation.  

Figure 1: Later figures list N=4 donor samples used for CD14+/CD14- population isolation. Where did those 4 samples fit in with this schematic? Were they part of the N=8 used in the digestion comparison?  

142: If the isolated cells were cultured for 10 days afterwards, how do you control for the fact that the primary synovial fibroblasts are expected to expand whereas the synovial macrophages do not? Or is there evidence that both populations will be maintained at the same ratios?  

146-147: From which strategy were the CD14 beads used? NDF/SDF/SDC or only downstream of the previously described protocol in [20]? Where does this fit into the Figure 1 schematic?  

150-164: What was the gating procedure for the flow cytometry data? Was a live/dead reagent used to exclude cells? Why were all markers evaluated with single colour rather than a multi-colour panel? This seems important for identifying macrophage expression of both M1- and M2-like surface markers. 

158-159: Were all synovial cells harvested from culture by washes with flow cytometry buffer without enzymatic or mechanical lifting of adherent cells?  

167: Was the 106 number of cells plated after the 7-day culture listed in line 140? Were the same cells used for flow cytometry and cytokine release?  

193: Kruskal-Wallis Test assumes independent observations between groups. Given that comparing NDF, SDF, and SDC should be repeated measures on donor-matched tissue, can this be accounted for in the statistical analysis?  

200-203: The experimental timeline is unclear from the Methods and Results sections. If the TKR tissue was evaluated by histology to identify the patients with moderate-grade synovitis, was the synovial tissue not processed with the three NDF/SDF/SDC protocols until the tissue was paraffinized, sectioned, stained, and graded? How would this be known at the time of receiving the synovial tissue sample? Were the low-grade synovitis samples discarded although culture was already started? The workflow should be clarified in the Figure 1 schematic.   

Figure 2: Labels for the ‘lining’ and ‘sublining’ layers discussed in the body text would be useful in your representative figures. Arrows highlighting ‘M1-like’ and ‘M2-like’ macrophages in the CD86/CD206 figure would be useful as the red/brown separation is hard to identify.  

Figure 2: What is the scale bar representative of in the high magnification image shown in F?  

222-224: Is there a reason why the lining/sublining distribution of M1-like and M2-like macrophages could not be quantified instead of described qualitatively? Were there any cells that were double positive for both the M1-like CD86 and M2-like CD206 markers or were all cells definitively single-stained as an M1 or M2 macrophage?  

226-229: It would be useful to report the mean and SD (or other summary statistics such as median/max/min) here. Is the percentage of fibroblasts and macrophages additive to 100%? If not, what is the expected % of cells that are not of those two populations?  

227-229: It is not clear what the purpose of doing a statistical comparison test between CD68 and CD86 or CD206 positive cells as if they are three independent populations. There is no takeaway conclusion from this observation. Assuming CD68 is the pan-macrophage marker, it would be inclusive of any cells expressing CD86 or CD206. Do the percentages of CD86+ and CD206+ cells add up to the percentage of CD68+ cells? Would that imply that CD86+ and CD206+ cells are mutually exclusive? And if not, then what is the population of CD86+/CD206+ double positive cells? 

233: Why were these four fibroblast markers selected for synovial tissue? What does it imply when they all differ in % positive cells in Figure 3? Are there different subpopulations of fibroblasts that should be important in OA, and are they being retained in the digestion process?  

234: Why is CD14 used here as the ‘synovial macrophage marker’ when CD68 is used as well? Explain the difference between the two markers and why they are both needed in your analysis. Why was CD68 used in IHC and not CD14? From Figure 3, why are the percentages for CD14+ and CD68+ different when they should be the same cell population? If not, are there differences in the macrophages that are CD14+/CD68- or vice versa?  

236: CD45 is noted here to have a very low percentage. Do synovial macrophages not express CD45? Other resident macrophage populations have varying levels of low-to-high expression of CD45 but still should appear positive by flow cytometry, as they are from a hematopoietic lineage.   

Figure 3: Some markers show significant variation in % positive cells, particularly in F. How do you support that this protocol will produce less variable results over explant tissue culture?  

251-256: The results described here demonstrate that all M1-like and M2-like macrophage markers are significantly higher in the NDF process. How does this relate to reproducing an OA-like environment in which to test OA therapies? If all the markers are elevated, is the M1/M2 macrophage ratio from the in vivo environment maintained in this culture? Is the M1/M2 macrophage ratio from these flow results comparable to the M1/M2 ratio from the IHC double staining data?  

Figure 4: What about the mean fluorescence intensity of these macrophage markers? M1-like and M2-like markers are not absolutes, especially from primary tissue. Macrophages could have surface expression of both M1 and M2 markers but express one phenotype with a significantly higher intensity.  

Figure 4: CD80/CD86 represents one population together, why are the % positive cells different for the two markers? Similarly, what about the discrepancy in % positive cells for CD163 and CD206?  

Figure 6: With a sample size of n=8, it may be more valuable to express the data here as scatter plots rather than box plots, particularly in the case of IL-10 (pg/mL) to communicate true values.  

291: Header is confusing. Not sure if the takeaway is that there are lower amounts of all measured secreted factors in both the CD14+ and CD14- populations?  

294: From which samples were the n=4 taken? Was CD14 isolation done separately on NDF, SDF, and SDC samples or was it pooled?  

295: From Figure 3, flow cytometry shows that there is <20% of CD14-positive cells from the NDF, SDF, and SDC samples versus >20% of CD68-positive cells. Thus, there must be a population of cells that is CD14-negative but CD68-positive. Are these synovial macrophages? Are these macrophages not isolated into the CD14-positive population? Are they included in the CD14-negative population and thus producing the cytokines and chemokines?  

334-335: Please clarify why only moderate-grade synovitis samples were chosen. Does this imply that the OA therapies to be tested with the synovial cells will only work on a certain OA population with moderate-grade synovitis? Can the digestion protocol and treatment be used on patient samples with low-grade synovitis to test future therapies? Is this a limitation?  

339: The conclusion drawn here is that none of the treatments had a specific impact on macrophages, defined by CD14 percentage positive. However, Figure 2 clearly outlines a difference between the treatments for CD68, which was claimed to be a synovial macrophage marker in Figure 2 and in the body text. 

343: If making the claim that the fibroblast/macrophage percentages are similar between the IHC results and flow cytometry results, this would have worked better as a single figure, or if numerical values were provided. Why not compare this in a table or plot? Why not use a statistical comparison test here to show the %positive fibroblasts and %positive macrophages is similar between the IHC samples and isolated cell samples?  

343: Can you compare the IHC samples matched to the donor isolated cell cultures?  

352-354: Your claim here is that cell subpopulations were lost due to the focus on adherent cells and not suspension cells. Are there other reasons why the in vitro culture favoured the growth of some cells over others? It was not clear why OPTIMEM was selected as the culture media. This is important to explain if the objective of this paper is to provide an isolation procedure that should be adopted by the OA research community.  

354: The argument made here is confusing. Enzymatic digestion is known to influence surface marker detection due to protein cleavage. However, given that your treatments after digestion have a 7-day culture period, removal of tissue fragments, and subsequent 10-day culture, would the cells not have ample time to reproduce the cleaved surface markers? Even if the markers are temporarily cleaved to demonstrate a difference in fresh PBMCs, the cells themselves still hold their identity despite not being detectable by surface markers immediately after enzymatic digestion. A different reference to literature here would be required to make the argument that enzymatic cleavage at time of isolation will affect viability or cell behaviour in subsequent in vitro culture.  

366-367: Due to the assay methods used here, it’s not clear that there were two distinct macrophage subpopulations or if macrophages expressed both M1 and M2 markers. Further, in Supplementary Figure 1, it appears that many CD14+ cells are neither CD206+ or CD86+. What about the macrophage population that is not expressing M1-like or M2-markers?  

389-390: This restates what was said in the results. Can you elaborate on why the molecular level observations were not recapitulated at the protein level?  You provide a reference that IL-10 is produced by cell populations not found in your isolate, but this does not corroborate with the fact that IL10 expression was present by gene expression. There was no discussion about the lack of correlation in the TGF-B1 protein level.  

403-405: This cited literature appears to claim that CCL18 is correlated with disease worsening. Does this imply that M2-like macrophages are detrimental in OA? Are you maintaining the correct M2-like behaviour and phenotype in culture? 

409: This paragraph discusses that synovial explants have a limited number. The experiments in this manuscript do not demonstrate the advantage of the digestion/isolation methods over explant culture. This would be better supported if cell number yields and viability values were provided for readers to visualize how this method would be better than explant culture to test in vitro OA drugs.  

Conclusion: Given that the introduction claims that M1/M2 ratio dysregulation (i.e. balance) impacts OA, then there should be more emphasis on maintaining the OA disease ratio of M1/M2 or whatever the original ratio was for each donor. The current data analysis supports that the NDF treatment methods retains more macrophages of both M1 and M2 only.  

Author Response

REVIEWER 1

There seems to be a lot of data that was collected for this study, but a lot of the analysis did not support the conclusions made. Results and discussion should focus on a more logical step-by-step to break down the synovial cell populations observed. There is a interchangeable use of what constitutes as your synovial fibroblast and synovial macrophage markers without ample justification, particularly where two markers are both claimed as well-known markers, but do not represent positive populations of each other.  

Line-specific comments: 

Point 1:68-71: Some of these references are not specific to OA. Is this of concern? Other arthritic conditions have different manifestations of synovium inflammation and tissue condition that may call for different isolation techniques.  

Response 1: We thank the reviewer for this observation. Synovial inflammation is an important feature of different arthritis diseases and only a few manuscripts are focused on OA disease (27, 28).However, even if different synovial isolation techniques may be required for other arthritic conditions, we only wanted to evidence that different procedures/protocols are used for synovial evaluation. Some studies are focused on synovial biopsy, other on isolated synovial cells and other on cell lines. The different procedures/protocols used could have influenced the results obtained. We have modified the sentence as follow:” However, the protocols used in the different studies could have positively or negatively influenced the results obtained.” (lines 80-81)

Point 2: 71: Multiple publications are cited here. What kind of variation was seen in these protocols and what made the variation ‘significant’?  

Response 2: As indicated by the reviewer, we have checked the references and the sentence also according to the response 1. In order to be more clear, we have modified the sentence as follows:” However, the protocols used in the different studies could have positively or negatively influenced the results obtained.” (lines 80-81)

Point 3: 81-84: Have you completed validation experiments to support the claim that the isolated cell populations can be used to test new therapies for OA? The current data presented does not show if the three different isolation protocols produce cells that respond differentially to known pro- or anti-inflammatory stimuli. Perhaps further elaboration or citation of previous work would be sufficient.  

Response 3: We thank the reviewer for this observation. In our previous works (62, 10) we have tested the effects of adipose-derived mesenchymal stromal cells both on inflamed and non-inflamed isolated synovial cells. These references are now included in the Discussion to support the claim that this model can be used to test new therapies for OA (lines765-767).

Point 4: 95-96: How were ‘inflamed characteristics’ selected from the synovial tissue specimens for fixation? Is this done by the surgeon or the lab operator? What characteristics defined this selection before Krenn scoring was done on the H&E samples?  

Response 4: OA patients undergoing total knee replacement were selected on the bases of clinical and radiological examination. Synovial tissues collected from each OA patient was evaluated by trained lab operators that selected only areas with specific macroscopic signs like hypervascularity and hyperplasia. From each synovial tissue we selected approximately 10-15 areas and, 2 of them were fixed for Krenn scoring while the others (approximately 13 areas) were used for isolation protocols. These details were now inserted into the manuscript (lines97-98, 141-143, 179-180).

Point 5: Figure 1: This current experimental schematic is unclear. Was the mechanical separation completed on the enzymatic digestion portion? In the Methods, the tissue was described to be cut into small fragments before splitting into ‘mechanical separation’ vs ‘enzymatic digestion’, but the schematic appears to place the scissors only in the enzymatic digestion route.  

Response 5: As suggested by the reviewer, we have now corrected Figure 1 and legend as described in the Methods (lines 177-222).

Point 6: Figure 1: The 7-day culture of the fragments in petri dishes and subsequent 10-day culture should be outlined in this schematic before ‘evaluation performed’. Please clarify if all evaluation and harvest of the cell populations was only done after a total of 17 days incubation.  

Response 6: As requested by the reviewer we have inserted in Figure 1 the experimental time point and corrected the sentence in the Methods (lines 201-203).

Point 7: Figure 1: Later figures list N=4 donor samples used for CD14+/CD14- population isolation. Where did those 4 samples fit in with this schematic? Were they part of the N=8 used in the digestion comparison?  

Response 7:Assuggested by the reviewer, we have inserted CD14+/CD14- population isolation in Figure 1. The four samples were part of the 8 samples used for comparing isolation procedures. These details are now inserted in Figure 1 and the Methods (lines 220-222).

Point 8: 142: If the isolated cells were cultured for 10 days afterwards, how do you control for the fact that the primary synovial fibroblasts are expected to expand whereas the synovial macrophages do not? Or is there evidence that both populations will be maintained at the same ratios?  

Response 8: To better compare the two procedures we performed all the evaluations on cells at passage 0 (lines 200-202). In our previous work (10) we compared different cell passages evidencing a decrease of macrophages from passage 0 to passage 2 and absence of these cells at passage 5. For this reason, we selected the passage 0 to compare both procedures. Moreover, we firstly analysed the percentage of positive CD55, CD68, CD80 and CD206 (Figure 2) on fixed synovial tissue specimens and then we checked the same markers also on day 10 cultured cells (Figure 3 and 4) to assure that the populations ratio was still maintained in similar percentages.

Point 9: 146-147: From which strategy were the CD14 beads used? NDF/SDF/SDC or only downstream of the previously described protocol in [20]? Where does this fit into the Figure 1 schematic?  

Response 9: As suggested by the reviewer we have now indicated in Figure 1 that we isolated CD14 positive and negative cells from SDF isolated cells as indicated in the Methods (lines 220-223) and as we previously published (15).

Point 10: 150-164: What was the gating procedure for the flow cytometry data? Was a live/dead reagent used to exclude cells? Why were all markers evaluated with single colour rather than a multi-colour panel? This seems important for identifying macrophage expression of both M1- and M2-like surface markers. 

Response 10: The cells were firstly counted by exclusion dye (eosin) before flow cytometry analysis and we found that they were 99% live (lines 248-250) and these cells were the ones gated. We performed single marker quantitative analysis on synovial tissue, so we decided to use the same procedure also on isolated cells using the same antibodies.

Point 11: 158-159: Were all synovial cells harvested from culture by washes with flow cytometry buffer without enzymatic or mechanical lifting of adherent cells?  

Response 11: We have clarified that for the flow cytometry analysis, cells were detached, counted and, after immunofluorescent staining were analysed. We have now inserted these details into the manuscript (lines 248-250).

Point 12: 167: Was the 106 number of cells plated after the 7-day culture listed in line 140? Were the same cells used for flow cytometry and cytokine release?  

Response 12: As suggested by the reviewer, we have specified all evaluations were performed after 10-days culture (lines 201-203).

Point 13: 193: Kruskal-Wallis Test assumes independent observations between groups. Given that comparing NDF, SDF, and SDC should be repeated measures on donor-matched tissue, can this be accounted for in the statistical analysis?  

Response 13: We thank the reviewer for this observation. We have used for statistical analysis the Friedman test and Dunn’s post hoc test for comparing NDF, SDF, and SDC, but we realized that we have indicated the wrong test. We have now performed the correction in the statistical analysis section (Figure legend 2). Moreover, we have specified that we have used the Mann Whitney U test for comparing CD14 positive versus CD14 negative cells (lines 290-292 and Figure legends 7 and 8).

Point 14: 200-203: The experimental timeline is unclear from the Methods and Results sections. If the TKR tissue was evaluated by histology to identify the patients with moderate-grade synovitis, was the synovial tissue not processed with the three NDF/SDF/SDC protocols until the tissue was paraffinized, sectioned, stained, and graded? How would this be known at the time of receiving the synovial tissue sample? Were the low-grade synovitis samples discarded although culture was already started? The workflow should be clarified in the Figure 1 schematic.   

Response 14: We thank and understand the observation of the reviewer. As described in point 4, synovial sample specimens obtained from each OA patient were evaluated by trained lab operators that selected approximately 10-15 areas with specific macroscopic signs of hypervascularity and hyperplasia. Then, 2 of the selected areas were fixed for Krenn scoring and the others were immediately used for isolation protocols. The isolated cells were then included in the study only if the Krenn score was between 4-6 (moderate grade). Following this procedure, the low-grade synovitis samples were discarded. We have now inserted these details into Figure 1 legend (206-219) and Methods (lines97-99, 141-143, 179-181, 201-203).

Point 15: Figure 2: Labels for the ‘lining’ and ‘sublining’ layers discussed in the body text would be useful in your representative figures. Arrows highlighting ‘M1-like’ and ‘M2-like’ macrophages in the CD86/CD206 figure would be useful as the red/brown separation is hard to identify.  

Response 15: As suggested by the reviewer we have now inserted in Figure 2 and figure legend the labels and arrows requested (lines 320-337).

Point 16: Figure 2: What is the scale bar representative of in the high magnification image shown in F?  

Response 16: We have inserted in the legend of Figure 2 the µm corresponding to the bar in image F (line 336).

Point 17: 222-224: Is there a reason why the lining/sublining distribution of M1-like and M2-like macrophages could not be quantified instead of described qualitatively? Were there any cells that were double positive for both the M1-like CD86 and M2-like CD206 markers or were all cells definitively single-stained as an M1 or M2 macrophage?  

Response 17: We thank the reviewer for this observation. We have chosen to quantify only single M1 or M2 stained cells since we only observed a few double positive stained CD86/CD206 cells.

Point 18: 226-229: It would be useful to report the mean and SD (or other summary statistics such as median/max/min) here. Is the percentage of fibroblasts and macrophages additive to 100%? If not, what is the expected % of cells that are not of those two populations?  

Response 18: We have now inserted the mean and SD of single analysed markers (360-362). As reported by other studies (1,5), the percentage of cells that are not fibroblasts or macrophages in OA synovial tissue is approximately 5-10%. In this study the sum percentage of synovial fibroblasts and macrophages calculated on synovial tissues is approximately 90% as we also previously reported (10).

Point 19: 227-229: It is not clear what the purpose of doing a statistical comparison test between CD68 and CD86 or CD206 positive cells as if they are three independent populations. There is no takeaway conclusion from this observation. Assuming CD68 is the pan-macrophage marker, it would be inclusive of any cells expressing CD86 or CD206. Do the percentages of CD86+ and CD206+ cells add up to the percentage of CD68+ cells? Would that imply that CD86+ and CD206+ cells are mutually exclusive? And if not, then what is the population of CD86+/CD206+ double positive cells?

Response 19: We thank the reviewer for this observation. As indicated, CD68 is the pan-macrophage marker inclusive of macrophages expressing CD86 or CD206 so it is clearly significantly higher. We have now removed the statistical analysis on both Figure and legend 2G.

Point 20: 233: Why were these four fibroblast markers selected for synovial tissue? What does it imply when they all differ in % positive cells in Figure 3? Are there different subpopulations of fibroblasts that should be important in OA, and are they being retained in the digestion process?  

Response 20: We selected these four markers also considering references (37-39) that demonstrated that OA synovial fibroblasts are mesenchymal cells that display many characteristics of fibroblasts. We have inserted these references in the manuscript (line359).

Point 21: 234: Why is CD14 used here as the ‘synovial macrophage marker’ when CD68 is used as well? Explain the difference between the two markers and why they are both needed in your analysis. Why was CD68 used in IHC and not CD14? From Figure 3, why are the percentages for CD14+ and CD68+ different when they should be the same cell population? If not, are there differences in the macrophages that are CD14+/CD68- or vice versa?  

Response 21: We have used both monoclonal antibodies for CD14 and CD68 as synovial macrophage markers but, as suggested by the reviewer, we noted that the sentence was not clear. We have now modified as follows: “Interestingly, synovial macrophage markers CD14 showed the same percentage while CD68 was significantly higher in NDF and SDF compared to SDC (p=0.0424 and p=0.0183, respectively)” (lines 360-362). CD14 and CD68 recognize two different antigens that could explain the different percentages showed in Figure 3. In order to have a complete quantitative evaluation on synovial tissues of synovial macrophages and M1/M2 subpopulations, we decided to analyse by IHC only 3 (CD68, CD86, CD206) out of the 6 (CD14, CD68, CD 80, CD86, CD163, CD206) macrophage markers analysed. However, the main focus of the study was on isolation procedures characterization of macrophages and fibroblasts, but not on the evaluation of subpopulations (i.e. CD14+/CD68-), that could be surely a good suggestion for further studies.

Point 22: 236: CD45 is noted here to have a very low percentage. Do synovial macrophages not express CD45? Other resident macrophage populations have varying levels of low-to-high expression of CD45 but still should appear positive by flow cytometry, as they are from a hematopoietic lineage.   

Response 22:We thank the reviewer for this observation. The monoclonal anti-CD45 that we have used in our study recognizes the antigen CD45RA that, as reported in the literature (38), is negative on macrophages. We have now corrected in the text that we have used the monoclonal anti-CD45RA (lines 256 and 362).

Point 23: Figure 3: Some markers show significant variation in % positive cells, particularly in F. How do you support that this protocol will produce less variable results over explant tissue culture?  

Response 23: Synovial tissue in OA patients is known to show patched areas of inflammation where the percentage of macrophages could be significantly different (references 1-6-10-35). As suggested by the reviewer we have integrated the strength of our study in the discussion as follows: “By contrast, the strength of this study is based on the use of a mixed population of synovial fibroblasts and macrophages. This avoids the use of a restricted number of synovial explants that do not adequately represent the overall composition of the OA synovial tissue (characterized by patched inflamed areas), and limits the evaluation of different experimental points” (lines 684-685, 756-762). Moreover, as we reported in our previous report (62) synovial moderate grade of inflammation is mainly characterized by an active release of IL6 (showing a median values of 3192 pg/ml associated to wide minimum-maximum values) that is fundamental for evidencing the positive effect of OA based cell therapies.

Point 24: 251-256: The results described here demonstrate that all M1-like and M2-like macrophage markers are significantly higher in the NDF process. How does this relate to reproducing an OA-like environment in which to test OA therapies? If all the markers are elevated, is the M1/M2 macrophage ratio from the in vivo environment maintained in this culture? Is the M1/M2 macrophage ratio from these flow results comparable to the M1/M2 ratio from the IHC double staining data?  

Response 24: The ratio of M1/M2 from IHC is approximately equal to 1. The analysis of CD80/CD163 and CD86/CD206 ratios in NDF isolated cells shows that they were approximately equal to 1 indicating that the in vivo environment is well maintained in cell culture. This observation is now included in the conclusion as follows:“NDF isolation procedure can retain synovial fibroblasts and a high ratio of M1/M2-like macrophages, as we found in synovial tissue, suggesting that an OA-like environment was maintained in the in vitro culture, and providing new indications and future directions for synovial OA research in vitro models”.(lines 771-773)

Point 25: Figure 4: What about the mean fluorescence intensity of these macrophage markers? M1-like and M2-like markers are not absolutes, especially from primary tissue. Macrophages could have surface expression of both M1 and M2 markers but express one phenotype with a significantly higher intensity.  

Response 25: We thank the reviewer for this observation. We have now inserted in the text the legend of Figure 4 and in the Figure 4 the mean fluorescence intensity (MFI) of all these macrophage markers. These new data confirmed a higher intensity of these markers on NDF isolated cells compared to SDC (lines 418-421).

Point 26: Figure 4: CD80/CD86 represents one population together, why are the % positive cells different for the two markers? Similarly, what about the discrepancy in % positive cells for CD163 and CD206?  

Response 26:As reported (11-13 and Vogel D Immunobiology  2014 Sep;219(9):695-703) the identification of M1 and M2 macrophages requires the use of at least two different markers as we have done in the study using CD80/86 for M1 and CD163/CD206 for M2 macrophages. These markers recognize different antigens present on M1 or M2 macrophages and for this reason are probably not exactly expressed in the same percentage. 

Point 27: Figure 6: With a sample size of n=8, it may be more valuable to express the data here as scatter plots rather than box plots, particularly in the case of IL-10 (pg/mL) to communicate true values.  

Response 27: We thank the reviewer for this observation. We decided to present all data as box-plot with minimum and maximum values to show a more clear distribution of the data. However, to clarify the reason of our decision to use box-plot, we have prepared the scatter plot of IL10 that does not show well, a clear distribution of the data into the three groups.

Point 28: 291: Header is confusing. Not sure if the takeaway is that there are lower amounts of all measured secreted factors in both the CD14+ and CD14- populations?  

Response 28: As suggested by the reviewer, we have changed the header as follows: Lower amounts of all measured secreted factors in both the CD14 positive and CD14 negative populations (lines 486-487).

Point 29: 294: From which samples were the n=4 taken? Was CD14 isolation done separately on NDF, SDF, and SDC samples or was it pooled?  

Response 29: As we have indicated in response 7, the four donor samples are part of the 8 used for comparing isolation procedures. From 4 OA cases we collected a higher number of synovial sample specimens (approximately 20 instead of 10-15) that permitted us to isolate CD14 positive and negative cells. As indicated in our previous work (15) and in the Materials and Methods section the CD14 cells were separated from digested synovial fragments (SDF) (lines 220-222).

Point 30: 295: From Figure 3, flow cytometry shows that there is <20% of CD14-positive cells from the NDF, SDF, and SDC samples versus >20% of CD68-positive cells. Thus, there must be a population of cells that is CD14-negative but CD68-positive. Are these synovial macrophages? Are these macrophages not isolated into the CD14-positive population? Are they included in the CD14-negative population and thus producing the cytokines and chemokines?  

Response 30: As indicated by the reviewer, in Figure 3 we reported that the percentage of CD14 positive macrophages was lower than CD68 and this result is surely due to the different antigen recognized by the two monoclonal antibodies on macrophage populations. CD14 positive cells are recognized as a synovial macrophage population and also other studies (Takano S. Clin Exp Immunol 2017 190: 235-243, 2017; Danks L et al Ann Rheum Dis 61:916-21, 2002) use this antibody for selecting synovial macrophages from fibroblasts. In fact, the CD14 negative cells are the fibroblast-rich fraction that, as we have shown in Figure S1, is not positive to other macrophage markers analysed (CD68, CD80, CD206).

Point 31: 334-335: Please clarify why only moderate-grade synovitis samples were chosen. Does this imply that the OA therapies to be tested with the synovial cells will only work on a certain OA population with moderate-grade synovitis? Can the digestion protocol and treatment be used on patient samples with low-grade synovitis to test future therapies? Is this a limitation?  

Response 31: We thank the reviewer for this observation. OA therapies such as cell therapies has been shown (Pers YM. et al Stem Cells Transl Med  2016 Jul;5(7):847-56.) to be effective in OA patients. Papers from our group (62) and from others (Desando G et al Arthritis Res Ther. 2013 Jan 29;15(1):R22; ter Huurne M et al. Arthritis Rheum. 2012 Nov;64(11):3604-13) have demonstrated that mesenchymal stromal cells for OA treatment require inflammation that triggers the activation of these cells and consequently determine their anti-inflammatory effects. Low-grade synovitis can be used to test future therapies, as we have previously reported (10) for cell therapies but, we do not know if this might be an advantage for other OA therapies.

Point 32: 339: The conclusion drawn here is that none of the treatments had a specific impact on macrophages, defined by CD14 percentage positive. However, Figure 2 clearly outlines a difference between the treatments for CD68, which was claimed to be a synovial macrophage marker in Figure 2 and in the body text. 

Response 32:We agree with the reviewer that it could be misleading to indicate CD14 positive macrophages considering that for CD68 we have evidenced a significant difference between NDF/SDF and SDC. Therefore, we have now modified this sentence as follows: “Our data demonstrated that the two synovial isolation procedures did not affect (at passage 0) the percentage of positive isolated cells on typical markers of synovial fibroblasts (CD55, CD73, CD90, CD106), suggesting that both mechanical or enzymatic treatments had no specific impact on surface markers, confirming a high percentage of synovial fibroblasts compared to macrophages (43). However, for typical macrophage markers we did not find differences using CD14 monoclonal antibody, but using CD68, CD80 or CD86 we found that the SDC procedure had a lower cell percentage than NDF or SDF.” (lines 554-560)

Point 33: 343: If making the claim that the fibroblast/macrophage percentages are similar between the IHC results and flow cytometry results, this would have worked better as a single figure, or if numerical values were provided. Why not compare this in a table or plot? Why not use a statistical comparison test here to show the %positive fibroblasts and %positive macrophages is similar between the IHC samples and isolated cell samples?  

Response 33:We thank the reviewer for this observation. As suggested, we have compared the data obtained with IHC and flow cytometry (considering the total cells isolated with the three procedures) and, as shown in the Figure below, we have not found significant differences and for this reason we have only reported that the percentage of isolated cells was “similar” to the percentage of cells analysed in the synovial tissue. (lines560-563)

Point 34: 343: Can you compare the IHC samples matched to the donor isolated cell cultures?  

Response 34: As we have reported in response 8, IHC samples and donor isolated cells derive from the same donor, 2  of the selected areas were used for IHC and all the others for cell isolation.

Point 35: 352-354: Your claim here is that cell subpopulations were lost due to the focus on adherent cells and not suspension cells. Are there other reasons why the in vitro culture favoured the growth of some cells over others? It was not clear why OPTIMEM was selected as the culture media. This is important to explain if the objective of this paper is to provide an isolation procedure that should be adopted by the OA research community.  

Response 35: As we have reported in lines 572-574 (reference 23, 26 and 27) the enzymatic digestion negatively impacted the isolation of mononuclear cells from synovial tissue. However, both our data and these studies have not considered the cells in suspension and we do not believe that other reasons could have favoured the growth of some cells over others. OPTIMEM medium was selected on the basis of preliminary experiments done in our Lab on synovial macrophages and fibroblasts that confirmed their typical phenotypic characteristics in vitro.

Point 36: 354: The argument made here is confusing. Enzymatic digestion is known to influence surface marker detection due to protein cleavage. However, given that your treatments after digestion have a 7-day culture period, removal of tissue fragments, and subsequent 10-day culture, would the cells not have ample time to reproduce the cleaved surface markers? Even if the markers are temporarily cleaved to demonstrate a difference in fresh PBMCs, the cells themselves still hold their identity despite not being detectable by surface markers immediately after enzymatic digestion. A different reference to literature here would be required to make the argument that enzymatic cleavage at time of isolation will affect viability or cell behaviour in subsequent in vitro culture.  

Response 36: We thank the reviewer for this observation. The FACS analysis was performed at day 10 of culture on enzymatic detached cells, so we have now modified the sentence as follows: “It has been shown on fresh PBMC (23) that enzymatic digestion influences the detection of immune cells markers, which could corroborate our results on retained immune adherent cells.” (lines 572-574)

Point 37: 366-367: Due to the assay methods used here, it’s not clear that there were two distinct macrophage subpopulations or if macrophages expressed both M1 and M2 markers. Further, in Supplementary Figure 1, it appears that many CD14+ cells are neither CD206+ or CD86+. What about the macrophage population that is not expressing M1-like or M2-markers?  

Response 37:We apologize since the sentence was not clear. We wanted only to underline that even when using mechanical and enzymatic digestion we did not find differences in the percentage of M1 and M2 synovial macrophages, however the enzymatic procedure could have the advantage of favouring the release of cells in a shorter time. We have now corrected the sentence as follows:” However, the enzymatic procedure (SDF) could have the advantage of shortening the time necessary for the release of cells in culture.” (lines 638-639). The evaluation of M1 (CD86+) - and M2 (CD206+) -like macrophages in synovial tissue evidences a similar percentage (approximately 12%) of these two cell populations. In supplementary Figure 1 we confirmed that CD14+ cells showed the same percentage of CD86 and CD206 as we found with IHC on synovial tissue. In supplementary Figure 1 we also showed that CD14- cells did not express macrophage markers CD68, CD86 and CD206.

Point 38: 389-390: This restates what was said in the results. Can you elaborate on why the molecular level observations were not recapitulated at the protein level?  You provide a reference that IL-10 is produced by cell populations not found in your isolate, but this does not corroborate with the fact that IL10 expression was present by gene expression. There was no discussion about the lack of correlation in the TGF-B1 protein level.  

Response 38:We thank the reviewer for this observation. It is interesting to note that both IL10 and TGFβ1 show the same expression in both CD14+ and CD14- cells, indicating a contribution of both synovial fibroblasts and macrophages both at molecular and protein levels. When synovial fibroblasts and macrophages were present in the same culture (as occurs in NDF, SDF and SDC) a cross-talk is established that could have determined a different modulation of IL10 and TGFβ1 at molecular level respect to protein level. It is well known that TGFβ1 and IL10 are expressed by both macrophages and fibroblasts and modulated during M2 polarization (12), suggesting that they could be differently accumulated at molecular levels while maintaining at protein level a sort of basal level that increases only under specific conditions of cell activation (like use of PGE2) or inhibition. We have now modified the statement as follows: “IL10 and TGFβ1…..(14). It is well known that TGF β1 and IL10 are both expressed by fibroblasts and macrophages (as we found for CD14+ and CD14-) and are modulated during M2 polarization (52,53). This suggests that these factors could be accumulated into the cells while maintaining a basal protein release and only specific conditions could modulate their production. In fact, it has been shown that high or low concentrations of TGFβ1 differently stimulate the balance among key signalling pathways (i.e. SMAD2/3 versus SMAD 1/5/8 (54, 55).” (lines 666-671)

Point 39: 403-405: This cited literature appears to claim that CCL18 is correlated with disease worsening. Does this imply that M2-like macrophages are detrimental in OA? Are you maintaining the correct M2-like behaviour and phenotype in culture? 

Response 39: We thank the reviewer for this observation. We have now modified the sentence as follows: “It has been shown that CCL18 is associated with OA disease severity and directly correlated with radiographic grading (60), suggesting that M2-like macrophages contribute to the evolution of the OA disease. Therefore, the maintenance of M2-like macrophages in culture is an important requisite to better mimic the OA milieu condition.” (lines 680-683)

Point 40: 409: This paragraph discusses that synovial explants have a limited number. The experiments in this manuscript do not demonstrate the advantage of the digestion/isolation methods over explant culture. This would be better supported if cell number yields and viability values were provided for readers to visualize how this method would be better than explant culture to test in vitro OA drugs.  

Response 40: As we have indicated in response 23, synovial tissue in OA patients is known to show patched areas of inflammation where the percentage of macrophages could be significantly different. By isolating the cells we homogenize the distribution in culture of the two main cell type fibroblasts and macrophages and at the same time we avoid the use of a limited number of synovial explants that do not adequately represent the overall composition of the OA synovial tissue. As reported in the Methods section from approximately 1200 mg/synovial fragmented sample we isolated 1.2 million viable cells. This cell number surely helps to perform the evaluation of different experimental points. We have now modified the sentence as follows: “By contrast, the strength of this study is based on the use of a mixed population of synovial fibroblasts and macrophages. This avoids the use of a restricted number of synovial explants, that do not adequately represent the overall composition of the OA ST (characterized by patched inflamed areas) and limits the evaluation at different experimental points. (lines 684-685, 755-758).

Point 41: Conclusion: Given that the introduction claims that M1/M2 ratio dysregulation (i.e. balance) impacts OA, then there should be more emphasis on maintaining the OA disease ratio of M1/M2 or whatever the original ratio was for each donor. The current data analysis supports that the NDF treatment methods retains more macrophages of both M1 and M2 only.  

Response 41: We thank the reviewer for this important indication. As suggested we have modified the conclusion as follows:“NDF isolation procedure well retain synovial fibroblasts and a high ratio of M1/M2-like macrophages, as we found in synovial tissue, suggesting that an OA-like environment was maintained in the in vitro culture, and providing new indications and future directions for synovial OA research in vitro models.” (lines 771-773)

Reviewer 2 Report

Manferdini and colleagues set out to create a suitable in vitro model of osteoarthritis with moderate-grade synovitis, in which both synovial fibroblasts and synovial macrophages are present and show behaviour characterisitic of these cells in vivo.

Such a model, they argue, is required to effectively explore the underlying pathophysiology and to identify important markers of disease progression. Alternative, currently used models tend to be too complex (tissue from knee surgery, which produce highly variable results) or too simplistic (monolayers or cells lines of single cell populations only, which may not adequately represent the more complex, diseased tissue). They suggest that a new model is required and provide a detailed evaluation of the methods they propose for this model, and the outcome in terms of the quality of the cells isolated and placed in culture.

The authors rated biopsies from patients undergoing knee replacement to determine the key level of synotivis and cellular composition they wished to focus on in the disease progression. These histological specimens were then used for molecular and protein studies of key characteristics of the affected cells. The authors are to be commended for using both methods to characterise the cells in passage 0, with the understanding that isolation and culturing may cause changes in cell phenotype.

Three methods were compared to the in vivo characteristics of oseoartritic tissue, synovial macrophages and synovial fibroblasts: mechanical isolation only (1) vs. mechanical isolation followed by enzymatic digestion - cells (2) and fragments (3) . While pro- and anti-inflammatory markers were frequently better expressed in cells cultured from mechanical isolation only, the results were not completely consistent between these three options.

The authors conclude that all three methods yielded mixed cultures with both synovial fibroblasts and macrophages (of both the M1 and M2 varieties), consistent with the in vivo pathophysiological condition for osteoarthritis with moderate synovitis. They recommend that their method be adopted for future studies of osteoarthritis with synovitis.

The studies are carefully designed and the cells carefully tested for how well the procedure preserved the expected cell characteristics. The authors are to be commended for testing whether the data were normally distributed, using the correct statistics and presentation of central tendency and variance when the distribution was skewed.

The authors are also to be commended for discussing a key limitation and key strengths of their study.

Major Points:

  1. The goal of the paper is to establish and characterise a novel and superior model of osteoarthritis with synovitis. However, the methods section did not, in my opinion, contain enough detail to recreate if I were to adopt their model. As a methods paper, I would argue that more detailed instructions are justified in the methods section. For example, what solution is used for mechanical digestion and subsequent enzymatic digestion - cell culture media, physiological saline, etc.? How were cells cultured on petri dishes? Were any extracellular matrix proteins used to coat culture dishes, to encourage cell adhesion? Such details are critical in recreating a new method.
  2. I understand that the development of a model with freshly cultured synovial fibroblasts and synovial macrophages is different from previous models/methods of studying osteoarthritis in vitro. However, were the results significantly different from that of monocultures or of less severe synovitis? In my opinion, the discussion would benefit from a critical assessment of how the proposed new model is superior to the currently used models.
  3. The authors conclude that all three versions of their in vitro model are suitable to study the role of synovial fibroblasts and synovial M1 and M2 macrophages in pathogenesis of osteoarthritis. They do not, however, rule in favour of one of these techniques over the other. Such a conclusion would be very helpful to future research, unless the authors propose that the combination of cultures from mechanical isolation with and without enzymatic digestion should all be employed to create the optimal model. In my experience, enzymatic digestion is more expensive and potentially damaging to cells, but generally creates higher yields of isolated cells. If mechanical isolation could be used without enzymatic digestion, surely that would be desirable? In my opinion, the discussion and conclusion would benefit if the authors would choose one of the three methods to put forward as the method to produce the optimal model.
  4. In the methods, the authors use a capital letter on the species name of Clostridium histoliticum (line 136). Please correct to be in keeping with the convention for taxonomic nomenclature.

Minor Points:

  1. Figure 1 has only a brief title. I would recommend a more elaborate description in the title, given that some future readers may ignore the text and focus on the figures preferentially. As the experimental plan (the method) is key in this paper, this figure is very important. It would be a pity if readers did not follow it properly, and a more elaborate figure description would help avoid this possibility.
  2. For subsequent figures, where p values are included in the figure, the values can be difficult to see as they run into the comparison bars. If the figures will be any smaller in the published version, I would recommend switching to the symbol (for example, "*" for p<0.05) convention to indicate the degree of significance. If the figures are at their final size, then would the authors consider making the font of the p values in the figure slightly smaller?
  3. Numerous grammatical errors are present. The authors are to be commended for writing in what may not be their mother tongue. However, some of the run-on sentences and issues with singular vs. plural make the prose difficult to understand in some places.

Author Response

REVIEWER 2

Manferdini and colleagues set out to create a suitable in vitro model of osteoarthritis with moderate-grade synovitis, in which both synovial fibroblasts and synovial macrophages are present and show behaviour characterisitic of these cells in vivo.

Such a model, they argue, is required to effectively explore the underlying pathophysiology and to identify important markers of disease progression. Alternative, currently used models tend to be too complex (tissue from knee surgery, which produce highly variable results) or too simplistic (monolayers or cells lines of single cell populations only, which may not adequately represent the more complex, diseased tissue). They suggest that a new model is required and provide a detailed evaluation of the methods they propose for this model, and the outcome in terms of the quality of the cells isolated and placed in culture.

The authors rated biopsies from patients undergoing knee replacement to determine the key level of synotivis and cellular composition they wished to focus on in the disease progression. These histological specimens were then used for molecular and protein studies of key characteristics of the affected cells. The authors are to be commended for using both methods to characterise the cells in passage 0, with the understanding that isolation and culturing may cause changes in cell phenotype.

Three methods were compared to the in vivo characteristics of oseoartritic tissue, synovial macrophages and synovial fibroblasts: mechanical isolation only (1) vs. mechanical isolation followed by enzymatic digestion - cells (2) and fragments (3) . While pro- and anti-inflammatory markers were frequently better expressed in cells cultured from mechanical isolation only, the results were not completely consistent between these three options.

The authors conclude that all three methods yielded mixed cultures with both synovial fibroblasts and macrophages (of both the M1 and M2 varieties), consistent with the in vivo pathophysiological condition for osteoarthritis with moderate synovitis. They recommend that their method be adopted for future studies of osteoarthritis with synovitis.

The studies are carefully designed and the cells carefully tested for how well the procedure preserved the expected cell characteristics. The authors are to be commended for testing whether the data were normally distributed, using the correct statistics and presentation of central tendency and variance when the distribution was skewed.

The authors are also to be commended for discussing a key limitation and key strengths of their study.

Major Points:

Point 1: The goal of the paper is to establish and characterise a novel and superior model of osteoarthritis with synovitis. However, the methods section did not, in my opinion, contain enough detail to recreate if I were to adopt their model. As a methods paper, I would argue that more detailed instructions are justified in the methods section. For example, what solution is used for mechanical digestion and subsequent enzymatic digestion - cell culture media, physiological saline, etc.? How were cells cultured on petri dishes? Were any extracellular matrix proteins used to coat culture dishes, to encourage cell adhesion? Such details are critical in recreating a new method.

Response 1:We thank the reviewer for these observations. We have now inserted more details in the Materials and Methods section and in Figure 1 to help the readers to reproduce the procedures described (lines 176-222).

Point 2: I understand that the development of a model with freshly cultured synovial fibroblasts and synovial macrophages is different from previous models/methods of studying osteoarthritis in vitro. However, were the results significantly different from that of monocultures or of less severe synovitis? In my opinion, the discussion would benefit from a critical assessment of how the proposed new model is superior to the currently used models.

Response 2: As suggested we have improved the discussion trying to better evidence the differences with monoculture and less severe conditions of synovitis. We have inserted in the discussion the following sentence:”We have previously demonstrated (10) that low inflamed cultures of synovial fibroblasts and macrophages are not efficient for testing cell OA therapies and also the use of synovial cell monoculture could represent a limit to reproduce in vitro an OA-like environment.” (lines 759-762)

Point 3: The authors conclude that all three versions of their in vitro model are suitable to study the role of synovial fibroblasts and synovial M1 and M2 macrophages in pathogenesis of osteoarthritis. They do not, however, rule in favour of one of these techniques over the other. Such a conclusion would be very helpful to future research, unless the authors propose that the combination of cultures from mechanical isolation with and without enzymatic digestion should all be employed to create the optimal model. In my experience, enzymatic digestion is more expensive and potentially damaging to cells, but generally creates higher yields of isolated cells. If mechanical isolation could be used without enzymatic digestion, surely that would be desirable? In my opinion, the discussion and conclusion would benefit if the authors would choose one of the three methods to put forward as the method to produce the optimal model.

Response 3: We thank the reviewer for this indication. We have now clearly stated that the NDF procedure is the best since it helps to retain in culture not only synovial fibroblasts but also more M1-like and M2-like macrophages (lines 771-773).

Point 4: In the methods, the authors use a capital letter on the species name of Clostridium histoliticum (line 136). Please correct to be in keeping with the convention for taxonomic nomenclature.

Response 4: We have now corrected following the convention for taxonomic nomenclature. (line 183)

Minor Points:

Point 5: Figure 1 has only a brief title. I would recommend a more elaborate description in the title, given that some future readers may ignore the text and focus on the figures preferentially. As the experimental plan (the method) is key in this paper, this figure is very important. It would be a pity if readers did not follow it properly, and a more elaborate figure description would help avoid this possibility.

Response 5: As suggested by the reviewer we have better described the legend in Figure 1, that now contains more details. (lines 206-219)

Point 6: For subsequent figures, where p values are included in the figure, the values can be difficult to see as they run into the comparison bars. If the figures will be any smaller in the published version, I would recommend switching to the symbol (for example, "*" for p<0.05) convention to indicate the degree of significance. If the figures are at their final size, then would the authors consider making the font of the p values in the figure slightly smaller?

Response 6: We thank the reviewer for this suggestion. We have now used in all the figures symbols for showing the significant difference between groups.

Point 7: Numerous grammatical errors are present. The authors are to be commended for writing in what may not be their mother tongue. However, some of the run-on sentences and issues with singular vs. plural make the prose difficult to understand in some places.

Response 7: As suggested by the reviewer, the manuscript was revised by an English mother tongue reviewer.
